# Human alveolar progenitors generate dual lineage bronchioalveolar organoids

Karen Hoffmann[1,13], Benedikt Obermayer [2,13], Katja Hönzke[1], Diana Fatykhova[1], Zeynep Demir [1], Anna Löwa[1], Luiz Gustavo Teixeira Alves [3], Emanuel Wyler [3], Elena Lopez-Rodriguez[4], Maren Mieth[1], Morris Baumgardt[1], Judith Hoppe[5], Theresa C. Firsching [5], Mario Tönnies[6], Torsten T. Bauer[6], Stephan Eggeling[7], Hong-Linh Tran[7], Paul Schneider[8], Jens Neudecker[9], Jens C. Rückert[9], Achim D. Gruber [5], Matthias Ochs [4,10], Markus Landthaler [3,11], Dieter Beule [2], Norbert Suttorp[1,10], Stefan Hippenstiel[1,10], Andreas C. Hocke[1,10] & Mirjana Kessler [1,12✉]

Mechanisms of epithelial renewal in the alveolar compartment remain incompletely understood. To this end, we aimed to characterize alveolar progenitors. Single-cell RNA-sequencing (scRNA-seq) analysis of the HTII-280⁺/EpCAM⁺ population from adult human lung revealed subclusters enriched for adult stem cell signature (ASCS) genes. We found that alveolar progenitors in organoid culture in vitro show phenotypic lineage plasticity as they can yield alveolar or bronchial cell-type progeny. The direction of the differentiation is dependent on the presence of the GSK-3β inhibitor, CHIR99021. By RNA-seq profiling of GSK-3β knockdown organoids we identified additional candidate target genes of the inhibitor, among others *FOXM1* and *EGF*. This gives evidence of Wnt pathway independent regulatory mechanisms of alveolar specification. Following influenza A virus (IAV) infection organoids showed a similar response as lung tissue explants which confirms their suitability for studies of sequelae of pathogen-host interaction.

[1] Department of Infectious Diseases and Respiratory Medicine, Charite-Universitätsmedizin Berlin, Corporate Member of Freie Universitat Berlin, Humboldt-Universität zu Berlin, and Berlin Institute of Health, Berlin, Germany. [2] Core Unit Bioinformatics, Berlin Institute of Health (BIH), Charité-Universitätsmedizin, Berlin, Germany. [3] Berlin Institute for Medical Systems Biology (BIMSB), Max Delbruck Center for Molecular Medicine in the Helmholtz Association (MDC), Berlin, Germany. [4] Institute of Functional Anatomy, Charité-Universitätsmedizin Berlin, Berlin, Germany. [5] Institute of Veterinary Pathology, Freie Universität Berlin, Berlin, Germany. [6] HELIOS Clinic Emil von Behring, Department of Pneumology and Department of Thoracic Surgery, Chest Hospital Heckeshorn, Berlin, Germany. [7] Department of Thoracic Surgery, Vivantes Clinics Neukölln, Berlin, Germany. [8] Department for General and Thoracic Surgery, DRK Clinics, Berlin, Germany. [9] Department of General, Visceral, Vascular and Thoracic Surgery, Charite-Universitätsmedizin Berlin, Corporate Member of Freie Universität Berlin, Humboldt-Universität zu Berlin, and Berlin Institute of Health, Berlin, Germany. [10] German Lung Center (DZL), Berlin, Germany. [11] IRI Life Sciences, Institute for Biology, Humboldt Universität zu Berlin, Berlin, Germany. [12] Present address: Department of Obstetrics and Gynecology, University Hospital LMU Munich, Munich, Germany. [13] These authors contributed equally: Karen Hoffmann, Benedikt Obermayer. ✉email: Mirjana.Kessler@med.uni-muenchen.de

Achieving substantial progress in the research of human lung diseases, from acute conditions to progressive chronic pathologies, largely depends on the establishment of suitable human in vitro models. Long-term organoid cultures derived from adult tissue capture in vitro renewal capacity of resident progenitor cells[1]. Adult stem cell (ASC)-derived organoid lines can be grown from a high number of parental tissues enabling the generation of biobanks[2–5]. These cultures are genetically stable[6] and show firm lineage commitment to the epithelial cell types of the organ they originate from. Stable expansion of airway organoids has been achieved by maintaining a specific niche environment containing Fibroblast growth factors FGF7 and FGF10, R-spondin1 (RSPO1), as well as blocking Bone morphogenetic protein (BMP) signaling, the Mitogen-activated protein (MAP) kinase pathway, and the Activin receptor-like ALK4/5 kinase, one of the Transforming growth factor-β (TGF-β) signaling routes[7,8]. These organoids were shown to be organized into basal, club, mucin-producing secretory, and ciliated cells and phenotypically match the global landscape of the airway epithelium gene expression as shown by bulk sequencing. Recently, the field has been advanced through the development of alveolar organoid cultures, which include the Glycogen synthase kinase-3 α and β (GSK-3α/β) inhibitor, CHIR99021 (short CHIR), in the growth medium[9,10]. GSK-3β is frequently referenced as the negative regulator of Wnt signaling due to its pivotal role in promoting beta-catenin degradation[11], but the kinase has also been shown to influence the intracellular signal transmission of Sonic Hedgehog (SHH)[12], and NOTCH[13] as well as PI3K/PTEN/mTOR1[14] cascade. Therefore, cellular mechanisms of the effect of CHIR on alveolar progenitors remain yet to be defined. In the mouse, multipotent (SCGB1A1+ SFTPC+) bronchioalveolar stem cells, termed BASCs, have been shown to regenerate both the bronchial and alveolar epithelium[15,16] but no human functional counterpart has been identified so far. By now proposed mechanisms of the regulation of the regeneration potential in the adult human lung all assume the presence of an independent pool of alveolar progenitors with only one unidirectional differentiation route leading from alveolar type 2 (AT2) precursors to alveolar type 1 (AT1) terminally differentiated cells[17–19]. Although extensive efforts have been made to identify subpopulations of the HTII-280+ AT2 epithelial cells that carry renewal potential, alveolar stem cells are yet to be identified[20]. Recent comprehensive study of alveolar organoids in co-culture with mesenchymal cells identified the ability of alveolar progenitors to transdifferentiate into basal KRT5+ airway cells, a phenotype, which is promoted in vivo by aberrant paracrine signaling in patients with idiopathic pulmonary fibrosis. It also confirmed potential of the inhibitor CHIR99021 to sustain alveolar lineage commitment in vitro[21].

Here, we wanted to know how the HTII-280+ epithelial compartment is organized, and how differentiation queues of the alveolar lineage are regulated, which are in vitro dependent on the presence of CHIR99021. By performing scRNA-seq analysis we have found that HTII-280+/EpCAM+ tissue cells uniformly express all markers of differentiated AT2 cells but different subclusters are identified based on the complete expression profile. Data analysis revealed positive enrichment scores for known adult stem cell signatures (ASCS)[22] of well-characterized epithelial tissue compartments in distinct subpopulations of HTII-280+/EpCAM+ cells, strongly suggesting the existence of intrinsic variability in the differentiation status and regeneration potential among AT2 cells. We observed high variability in the frequency of HTII-280+ cells in the peripheral lung potentially implying patient-dependent differences in renewal capacity. Importantly, we show that HTII-280+ progenitors have the potency to generate organoids with both alveolar and bronchial cell types. In

vitro induction and maintenance of an AT2 phenotype is dependent on the inhibition of GSK-3β, while the same progenitors in the medium without CHIR99021 give rise to organoids built of basal, secretory, and ciliated cells. Molecular action of CHIR99021 is more pleiotropic than specific inhibition of GSK-3β, as shRNA-mediated knockdown of GSK-3β in organoids does not phenocopy the action of the inhibitor. Rather, as identified by bulk RNA-seq of GSK-3β knockdown organoids, CHIR99021 independently regulates multiple signaling pathways, among which are potent regulators of quiescence, proliferation, and differentiation, like transcription factor *FOXM1* and growth factor *EGF*.

## Results

**HTII-280+ population contains clusters enriched for stem cell signature genes.** First, we sought to analyze the population diversity of HTII-280+ cells on the single-cell level upon isolation from human lung tissue. Although this surface membrane protein has been extensively used as a marker of the AT2 compartment, it remains unclear which subgroup within this lineage contains regeneration and longevity potential. In line with a previous study[23], confocal imaging of HTII-280+ cells in lung tissue explants confirmed consistent colocalization with SFTPC+ cells (Supplementary Fig. 1a). Accordingly, scRNA-seq of alveolar cells from peripheral lung tissue was performed with isolates from 6 donor lungs: 2 by sorting only HTII-280+ cells (Supplementary Fig. 8) and 4 by profiling HTII-280+/EpCAM+ cells to control for the fidelity of the epithelial phenotype. Analysis of a total of 33375 cells with a median count of 6949 UMIs and 1788 genes) (Supplementary Fig. 1b) revealed the nearly universal ubiquitous expression of surfactant genes *SFPTC*, *SFTPB*, and *SFTPA1* (Fig. 1a) after removal of likely contaminant clusters (Methods). Unsupervised cell clustering of epithelial cells identified 14 (0–13) different subpopulations (Fig. 1b) with both experimental setups (HTII-280+ and HTII-280+/EpCAM+) resulting in comparable subcluster composition (Supplementary Fig. 1c). This confirms that HTII-280 as a surface marker reliably identifies epithelial cells of the alveolar compartment. Among identified subclusters (Supplementary Fig. 1d and Supplementary Data 1) cells in cluster 4 co-express markers of differentiated AT1 cells (*AGER*, *EMP2*), which could be evidence of the continuous AT2 to AT1 transdifferentiation process in the tissue. Notably, while major markers of bronchial cell types remained at or below the detection limit (*KRT5*, *TP63*, *MUC5B*), the presence of a small cluster of *SCGB1A1* expressing cells (cluster 13) could indicate the existence of lineage mixed phenotypes in the human lung. To gain insight into the hierarchical organization of the alveolar compartment and identify candidate populations that confer stemness potential, we scored the adult stem cell signature (ASCS) developed previously by integrative pan-epithelial analysis of expression datasets from purified stem cells from the intestine and breast, as well as intestinal stem cells and their differentiated counterparts[22] (Supplementary Data 1). Within these and other adult stem cell niches, crosstalk between Wnt and BMP signaling has been identified as central for the control of stemness and differentiation potential. Thus, we tested whether similar mechanisms exist in the lung. Indeed, we detected high enrichment scores for ASCS in cluster 11 (Fig. 1c), as 8 genes from the signature set were found to be abundantly expressed in this cluster (Supplementary Data 1). Among markers of highest specificity for cluster 11, we have identified *FOXM1*, known to be involved in the regulation of quiescence and stemness potential in the hematopoietic compartment[24] and frequently dysregulated in malignancies (glioma, leukemia, and colon cancer)[25,26]. In mouse models, excessive activation of *FOXM1* in lung tissue has been associated

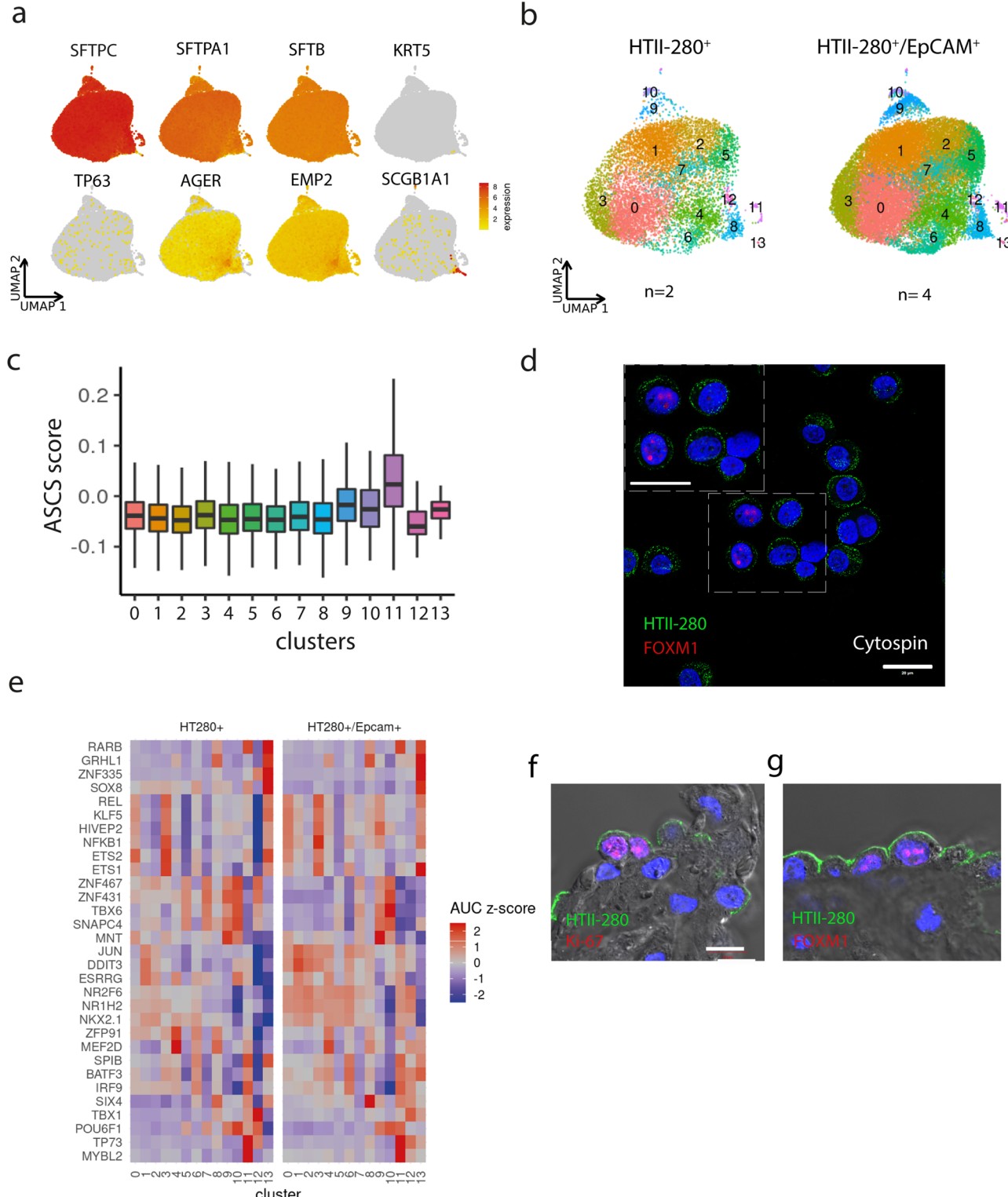

**Fig. 1 HTII-280$^+$ lung compartment contains distinct progenitor subpopulations. a** scRNA-seq of isolated HTII-280$^+$ cells shows that AT2 pneumocyte markers are ubiquitously expressed in contrast to bronchial lineage markers ($n = 6$ donors, 33375 cells). **b** Unsupervised clustering of HTII-280$^+$ and EpCAM$^+$/ HTII-280$^+$-derived scRNA-seq data **c** Gene signature scores for adult stem cell signature (ASCS). **d** Confocal image after cytospin of HTII-280$^+$ cells showing FOXM1 nuclear staining. Scale bar:20 μm. **e** SCENIC results of transcription factor regulons. **f** Confocal image representative of stainings from two different donor lungs showing proliferating AT2 cells (HTII-280 green, Ki67 red). **g** AT2 cells expressing FOXM1 transcription factor (HTII-280 green, FOXM1 red). Nuclei are counterstained with DAPI, and tissue is visualized by DIC. Scale bar: 20 μm.

with disruption of the regeneration potential and an increase in fibrosis[27]. Importantly, FOXM1 has been found to directly transcriptionally regulate the expression of the surfactant protein C (*SFTPC*) gene, which is a main marker of the AT2 alveolar phenotype and is essential for mouse lung maturation after birth[28]. To independently confirm the expression of FOXM1 in the subset of epithelial cells, HTII-280+/EpCAM+ cells from two different donor lungs were spun onto a glass slide using Cytospin and co-stained with anti-HTII-280 and anti-FOXM1 antibodies. Indeed a clear FOXM1 nuclear staining pattern was detected in a fraction of HTII-280+ cells as observed by confocal imaging (Fig. 1d). Next, we used SCENIC[29] to identify broader regulatory networks of transcriptional factors within the clusters (Fig. 1e). Again, the analysis revealed a large overlap between HTII-280+ and HTII-280+/EpCAM+ samples. Cluster 11 is found to be characterized by the activation of the MYBL2 network with notable enrichment also in the RARB pathway. As regulators of cell proliferation, FOXM1 and MYBL2 have been previously described as key players in the development of small lung cancer[30]. Thus, this data is suggestive of a putative involvement of these networks in regeneration and differentiation processes in adult lung. We concluded that HTII-280+/EpCAM+ cells contain functionally distinct subpopulations at different stages of differentiation, with cluster 11 likely comprising cells that harbor stemness potential. The presence of the candidate progenitor cells, actively proliferating Ki67+/HTII-280+ cells as well as HTII-280+ cells expressing FOXM1 in the nucleus, could also be confirmed in native lung tissue sections as visualized by confocal imaging (Fig. 1f, h) in agreement with the scRNA-seq data. To test the functional capacity of HTII-280+ cells in vitro we established a cohort of organoid lines under different experimental conditions.

The capacity of HTII-280+ epithelial cells to generate short-lived alveolar type organoids[31] was used as the basis for AT2 lineage selection by FACS upon which progenitors were seeded in 3D culture (Fig. 2a). Interestingly, the frequency of HTII-280+ cells in the primary tissue isolates varied considerably between donors (0.1–4.8%) potentially reflecting the influence of clinical background and comorbidities on the regeneration potential of the lung (Fig. 2b). The vast majority of samples were obtained as healthy tissue from lung cancer patients. To control for sample purity, we have performed CNV analysis on two representative lung tissue explants and six primary alveolar cell isolates using CONICS[32]. In order to detect potential subclones with CNV alterations, we grouped cells into 2 clusters based on CNV profiles derived from expression values averaged along chromosomes. However, the cells we have obtained in these clusters did not display any coherent localization in cancer-relevant cell types, indicating that these clusters likely do not exhibit genuine CNV aberrations but are defined rather by random expression fluctuations (Supplementary Fig. 2a). Besides underlying cancer diagnosis, the donor cohort group is characterized by a heterogeneous spectrum of secondary medical conditions (Fig. 2b). Thus, no clear correlation could be identified between available clinical data and the abundance of HTII-280+ cells[33,34]. To assess the differentiation potential and variability of the resulting phenotypes, HTII-280+ progenitors were cultivated in parallel in a medium optimized for airway stem cells in the presence and/or absence of CHIR99021. Indeed, 15/17 samples exhibited robust organoid forming efficiency in the medium containing CHIR ranging between 1–3% without correlation to the percentage of HTII-280+ cells in the parental tissue. Airway organoid lines generated in parallel from unselected pool isolates in airway organoid medium (AOM), on the contrary, showed a higher organoid formation and growth capacity compared to the HTII-280+ progenitors confirming the high expansion potential

of the airway stem cells (Fig. 2c). Interestingly, we found that HTII-280+ progenitors also generate expandable organoid lines in standard AOM without CHIR, albeit at lower efficiency (~0,3%) (Fig. 2d). Also, these organoids were smaller in size and with a lower expansion rate than the unselected pool organoids (passage time: 3 weeks vs 2 weeks, respectively), suggesting the existence of two separate progenitor populations for airway organoids. This was confirmed by parallel testing of the organoid forming efficiency of HTII-280+/EpCAM+ *versus* HTII-280−/EpCAM+ cells from 3 donor lung isolates in the AOM medium, which demonstrated that HTII-280−/EpCAM+ progenitors give rise to much larger and faster-growing airway organoids representing likely dominant drivers of airway regeneration (Fig. 2e). HTII-280+/EpCAM+ cells, on the contrary, represent a population that rather drives alveolar regeneration as alveolar organoids robustly grow in CHIR medium in agreement with previously published studies.

**Cell types within organoids reveal the complexity of differentiation processes in the lung.** The longevity of our alveolar organoid lines for >6 months in culture is ensured by the successful preservation of stemness potential in vitro, though we have observed growth arrest of donors 2 and 3 at 8–9 months, suggesting further optimization of the medium might be needed to achieve infinite growth in all cases. Notably, organoids grown from HTII-280+ sorted cells in the presence of CHIR displayed unique morphological features: overall smaller in diameter (average 50–200 μm), with loosely organized monolayer as illustrated by Hematoxylin and Eosin (HE) stain and phase-contrast images (Fig. 3a). Immunofluorescence analysis revealed a strong expression of SFTPC in these organoids, which is sustained during long-term cultivation (Fig. 3b; P3, ~3 months). Ultrastructural analysis of the epithelial layer by transmission electron microscopy showed that organoid cells display key properties of a fully differentiated AT2 phenotype (Fig. 3c). While the presence of tight junctions, adherens junctions, and apical microvilli are evidence of established structural integrity and polarity, the aggregation of secretory vesicles containing lamellar body-like structures in the organoid lumen strongly implies secretory activity, which is the main functional property of the surfactant producing AT2 cells in vivo. Induction of *SFTPC* and *HOPX*, a key regulator of AT2 to AT1 transition[35], was confirmed by qPCR in three independent donor lines (Supplementary Fig. 1b). Protein levels for selected AT2-specific markers in HTII-280+ organoids were also validated by immunoblot. SFTPC, NAPSIN A, and HOPX are detected in all alveolar organoid lines while being absent in the airway organoids of the same donor (Fig. 3d and Supplementary Fig. 7a). The stability of the phenotypes was confirmed by determining the relative expression of the SFTPC and HOPX during long-term cultivation (Supplementary Fig. 2c) for three independent donors.

To characterize the cellular composition and differentiation status of organoids from the alveolar induction protocol, scRNA-seq was performed of long-term cultivated HTII-280+-derived lines from 3 different donors (3, 5, and 6 months of in vitro expansion, total 7800 cells). Cluster annotation was performed based on reference data from the Human Lung Cell Atlas[36]. The analysis identified a large population of differentiated AT2 cells that constitutes between 7 and 45% of all cells within these organoids (Fig. 3e, f). Intermediate cell types encompass ~5% while the remaining cells are classified as bronchial types (basal and secretory). Joint embedding of 2000 organoid cells together with lung epithelium reference scRNA-seq data confirmed great similarity in expression profiles (Supplementary Fig. 3a) and SNV profiles showed only random variations (Supplementary Fig. 3b)

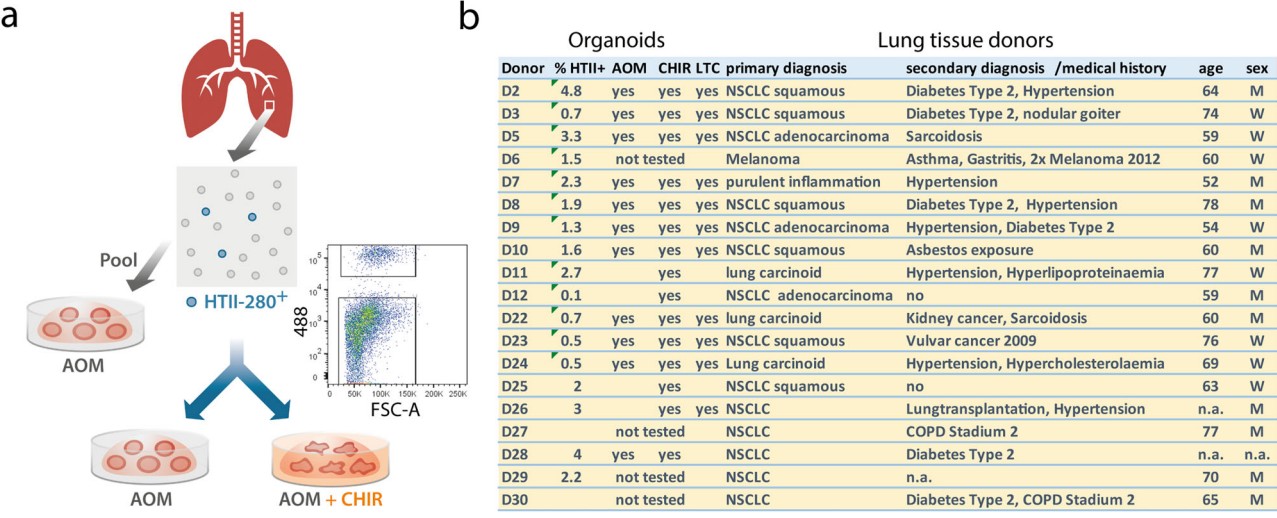

**a** Experimental layout

**b**

| Donor | % HTII+ | AOM | CHIR | LTC | primary diagnosis | secondary diagnosis /medical history | age | sex |
|---|---|---|---|---|---|---|---|---|
| D2 | 4.8 | yes | yes | yes | NSCLC squamous | Diabetes Type 2, Hypertension | 64 | M |
| D3 | 0.7 | yes | yes | yes | NSCLC squamous | Diabetes Type 2, nodular goiter | 74 | W |
| D5 | 3.3 | yes | yes | yes | NSCLC adenocarcinoma | Sarcoidosis | 59 | W |
| D6 | 1.5 | not tested | | | Melanoma | Asthma, Gastritis, 2x Melanoma 2012 | 60 | W |
| D7 | 2.3 | yes | yes | yes | purulent inflammation | Hypertension | 52 | M |
| D8 | 1.9 | yes | yes | yes | NSCLC squamous | Diabetes Type 2, Hypertension | 78 | M |
| D9 | 1.3 | yes | yes | yes | NSCLC adenocarcinoma | Hypertension, Diabetes Type 2 | 54 | W |
| D10 | 1.6 | yes | yes | yes | NSCLC squamous | Asbestos exposure | 60 | M |
| D11 | 2.7 | | yes | | lung carcinoid | Hypertension, Hyperlipoproteinaemia | 77 | W |
| D12 | 0.1 | | yes | | NSCLC adenocarcinoma | no | 59 | M |
| D22 | 0.7 | yes | yes | yes | lung carcinoid | Kidney cancer, Sarcoidosis | 60 | M |
| D23 | 0.5 | yes | yes | yes | NSCLC squamous | Vulvar cancer 2009 | 76 | W |
| D24 | 0.5 | yes | yes | yes | Lung carcinoid | Hypertension, Hypercholesterolaemia | 69 | W |
| D25 | 2 | | yes | | NSCLC squamous | no | 63 | W |
| D26 | 3 | | yes | yes | NSCLC | Lungtransplantation, Hypertension | n.a. | M |
| D27 | | not tested | | | NSCLC | COPD Stadium 2 | 77 | M |
| D28 | 4 | yes | yes | | NSCLC | Diabetes Type 2 | n.a. | n.a. |
| D29 | 2.2 | not tested | | | NSCLC | n.a. | 70 | M |
| D30 | | not tested | | | NSCLC | Diabetes Type 2, COPD Stadium 2 | 65 | M |

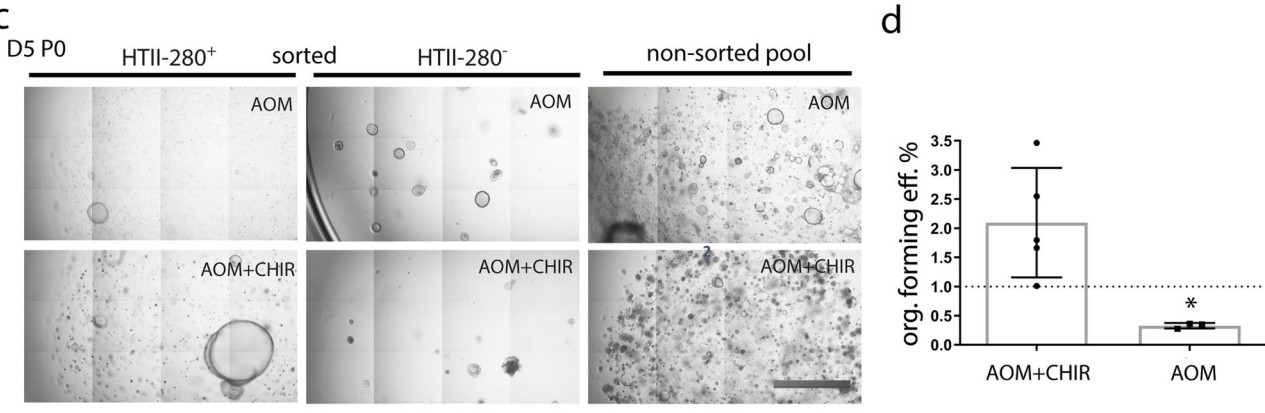

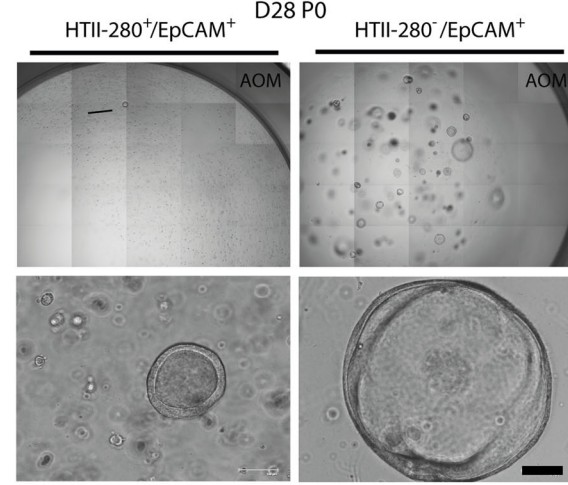

**Fig. 2 HTII-280+ progenitors from adult human lung generate in vitro organoids. a** Experimental layout of the progenitor isolation and generation of HTII-280+-derived organoid lines. **b** Summary of clinical data for donor lung tissue processed to generate organoid lines. LTC- long-term culture, n.a.—not available. **c** HTII-280+ cells give rise to organoids in AOM and AOM+CHIR medium, while HTII-280- epithelial progenitors have higher organoid forming efficiency in the AOM medium. Also, non-sorted pool isolates show robust growth in AOM medium. **d** Quantification of organoid forming efficiency of HTII-280+ progenitors in AOM + CHIR and AOM medium. $n = 5$ and $n = 3$, respectively, where $n$ are independent donor samples. Data represent mean +/−SEM. *$p = 0.02$ calculated by a two-tailed student $t$-test. **e** HTII-280-/EpCAM+ progenitors grow better in AOM medium compared to HTII-280+/EpCAM+ cells. The image is representative of three independent sort experiments. Scale bar: 100 μm.

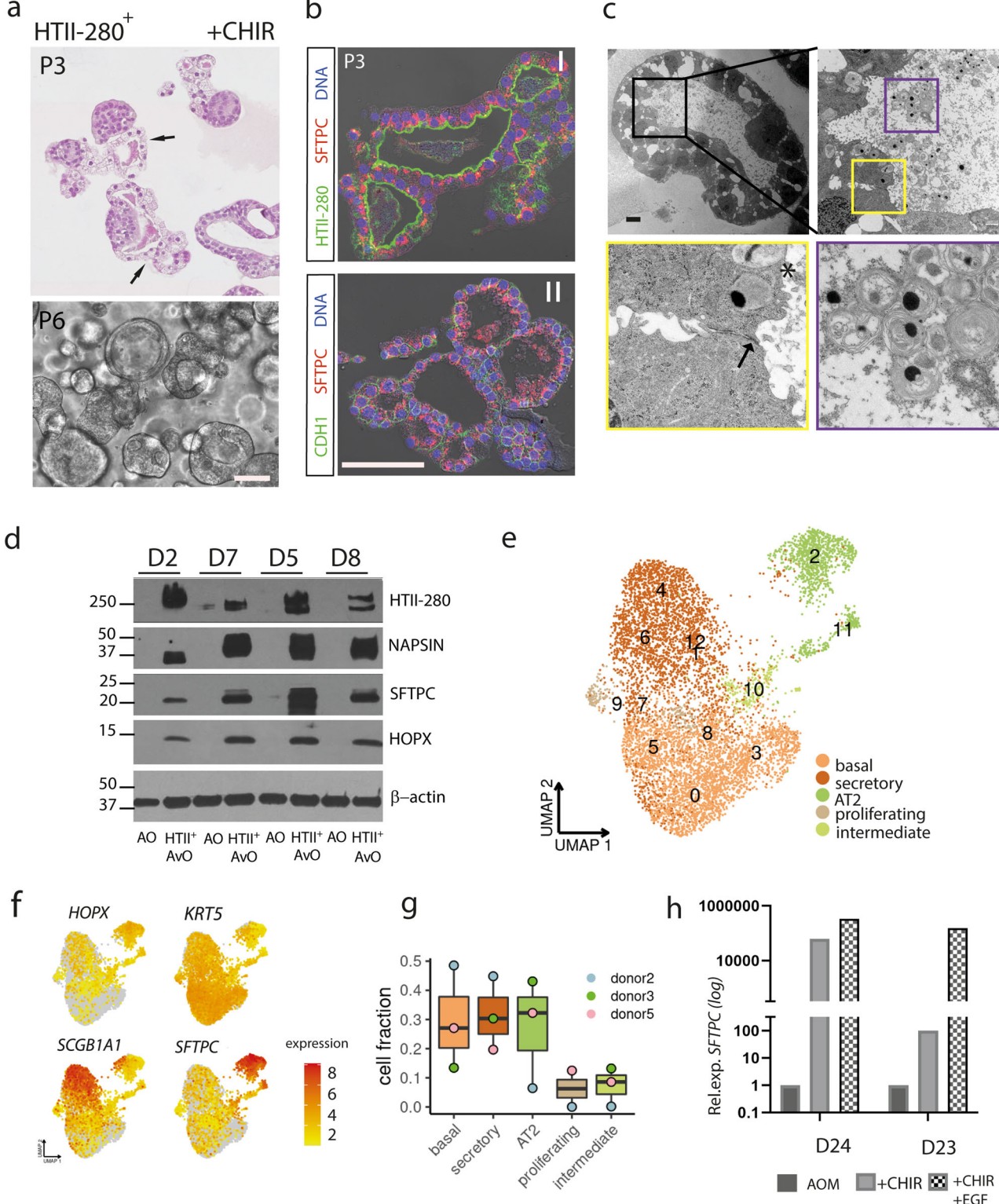

While some variation in the cell type proportions between the organoid lines remains, systematic batch effects have been removed in the analysis as evidenced by the even distribution of cells from different donors in the UMAP embedding (Supplementary Fig. 3c). Interestingly transcriptional profile of the alveolar cells is uniform among donors while other cell types show a considerable degree of individual variability (Supplementary Fig. 3d). Notably, though judged on the morphology and main structural features the organoids appear to contain

predominantly AT2 cells, the expression of bronchial markers like *KRT5* and *SCGB1A1* (Fig. 3e, f) in these lines is higher than previously reported by other groups who indicated that GSK-3β inhibition leads to the generation of pure expandable alveolar organoid cultures[9,10]. To assess the influence of EGF signaling on alveolar differentiation, used in other studies as a standard culture component, 50 ng/ml EGF was added to the CHIR growth medium. Indeed, qPCR analysis revealed that the *SFTPC* level does further increase in EGF-treated organoids in two different

**Fig. 3 HTII-280+-derived organoids maintain a stable AT2 phenotype in CHIR medium. a** HE staining and phase-contrast images showing the morphology of HTII-280+-derived organoids in CHIR medium. Scale bar: 100 μm. **b** Representative confocal images showing expression of the pneumocyte marker SFTPC (I, red) and maintained apical polarity (II, HTII-280, green,) in the long-term organoid culture (P3, ~3 months). Scale bar: 50 μm. **c** Representative transmission electron microscopy (TEM) images revealing fully formed junctions (arrow) and prominent microvilli (asterisk) of AT2 cells (yellow square). Abundantly secreted vesicles with lamellar membrane structures are present in the organoid lumen (violet square). Scale bars: 2.5 μm, 1 μm, 100 nm. **d** Protein levels of AT2 markers (SFTPC, NAPSIN A, HTII-280, and HOPX) in HTII-280+-derived alveolar organoids (AvO) compared to airway organoids (AO) of the same donor reveal sustained levels in all four lines, while no signal was detected in controls. **e** scRNA-seq analysis of alveolar organoids identifies clusters of AT2 cells, but also intermediate and airway cell types (secretory and basal cells) (n = 3 donors, 7784 cells). **f** Expression of HOPX, SFTPC, KRT5, and SCGB1A1 in the clusters derived from scRNA-seq of alveolar organoids. **g** Fractions of all identified cell types within the HTII-280+-derived organoids. **h** Relative expression level of SFTPC in HTII-280+-derived alveolar organoids treated with CHIR or CHIR+EGF relative to pool organoids grown in AOM for two different donors.

donor lines, which could explain somewhat less pronounced alveolar features in our standard culture conditions (Fig. 3h).

Nevertheless, because of the airway features detected in the expression profile of the HTII-280+-derived organoids cultured with CHIR we were interested to compare those to the phenotype of HTII-280+-derived organoids grown in a medium without the GSK-3β inhibitor. Imaging analysis of these organoids showed a clear shift in the cellular phenotypes and the levels of markers of alveolar and bronchial differentiation depending on the cultivation conditions. As shown in the composite of confocal images of organoids from two donor lungs AOM medium supports complete differentiation of bronchial phenotypes including the presence of ciliated cells and abundant subapical expression of SCGB1A1, evidence of prominent secretion of this protein to the lumen (Fig. 4a). In comparison, organoids cultivated in the alveolar medium with CHIR show strongly depleted levels of the club cell marker, and fully formed cilia are seldom observed. At the same time, ubiquitous levels of alveolar marker SFTPC were detected only in the presence of the inhibitor, in agreement with RNA expression levels of main differentiation markers as measured by qPCR in 3 pairs of lines from independent donors (Fig. 4b). Notably, bronchial lineage differentiation of HTII-280+ progenitors that form organoids in AOM is sustainable in vitro as the experiment was performed in passage 3 (2.5 months) confirming the stability of the diverging phenotypes in a favorable signaling environment. This finding strongly suggests a dual lineage potential of the HTII-280+ progenitors that is mediated in response to local exogenous niche factors. To determine whether in vitro formation of organoids permanently defines the direction of differentiation, HTII-280+-derived alveolar organoids were dissociated into single cells and subjected to HTII-280+ resorting with subsequently renewed seeding of HTII-280+-positive cells in 3D in AOM or AOM+CHIR medium (Fig. 4c). Resorted HTII-280+ cells preserved organoid forming capacity in 3D confirming again the stemness potential of this cell population. Reseeding of those progenitors in AOM triggered an immediate decrease in levels of alveolar marker NAPSIN A, while organoids in CHIR medium exhibited stable NAPSIN A protein levels but a clear reduction of TP63 (Fig. 4d and Supplementary Fig. 7b). We concluded that HTII-280+ progenitors have inherent lineage plasticity and assume alternative differentiation fates depending on the signaling environment. Moreover, long-term cultivation, followed over three passages (~2 months), caused a further increase in the levels of the airway (TP63, SCBG1A1) or pneumocyte markers (SFTPC, NAPSIN) in organoids grown in respective medium conditions (Supplementary Figs. 3e and 7c). Notably, HTII-280+-CHIR organoids did not exhibit significant upregulation of Wnt target genes AXIN2, LEF1, and TCF4, as measured by qPCR in comparison to HTII-280+-AOM counterparts with TCF4 even showing downregulation (Fig. 4e). We also detected significant downregulation of TGFB1 expression indicating the broader effect of CHIR beyond its influence on the Wnt pathway. To test the specificity of the CHIR99021 action, we

transduced alveolar organoids with lentiviruses carrying shRNA against GSK-3β (shGSK-3β), as well as a control vector carrying a non-mammalian shRNA (Fig. 5a). Interestingly, shGSK-3β progenitors failed to establish stable organoid growth in absence of CHIR99021, strongly suggesting that additional cellular targets of the inhibitor are responsible for the formation of alveolar organoids (Fig. 5b). In the medium containing CHIR99021 both control and shGSK-3β cells yielded stable expanding cultures. Knockdown efficiency was confirmed by qPCR and on protein level (Fig. 5c and Supplementary Figs. 3g and 7d). Considering the requirement for CHIR99021 action to support organoid formation irrespective of the presence of the main target GSK-3β, and to identify independently regulated genes, bulk RNA-sequencing was performed of shGSK-3β and control shRNA organoids. To ensure reliability and specificity of the knockdown effect, the experiment was performed with 2 different hairpins against GSK-3β. As CHIR is indispensable for organoid formation, the effect of the inhibitor was investigated by its removal from the medium at 5 days post-seeding and differential cultivation +/−CHIR for another week (Fig. 5a). Gene expression profiles of a total of 8 samples were then analyzed by bulk RNA-seq. Comparing gene expression changes upon CHIR withdrawal in both genetic backgrounds (Fig. 5d) revealed a substantial similarity (Pearson's R = 0.59) and very few genes with significant differences in their transcriptomic response (adj. p-value < 0.05) in line with the hypothesis that CHIR treatment has an effect on cellular targets beyond GSK-3β. Further, data confirmed that the presence of CHIR only weakly induces Wnt target genes (LEF1, WIF1) in control wild-type organoids (Fig. 5e). Thus, the alveolar differentiation process can not be interpreted as plane amplification of the Wnt signal. Accordingly, genes that show the same regulatory pattern upon CHIR withdrawal regardless of the level of GSK-3β could be involved in alveolar specification independent of Wnt signaling. Notably, FOXM1, found to be associated with the putative progenitor cluster within the native AT2 population, is also detected at higher levels in CHIR-treated organoids. Validation of relative mRNA expression of FOXM1 in HTII-280+-derived organoids cultivated in CHIR and AOM medium in two different donor lines confirmed the positive effect of the CHIR99201 treatment (Supplementary Fig. 3g). Interestingly, EGF expression was also found to be positively regulated in the CHIR medium in both genetic backgrounds. Importantly, it can be concluded that alveolar differentiation requires the action of CHIR99021 inhibitor functionally independent from its Wnt activating role, and we identify differentially expressed target genes that are putative candidates to mediate this process (Supplementary Data 2).

**AT2 pneumocytes disappear in pool organoids as only bronchial cells remain.** To test if the CHIR mode of action is the same on all epithelial progenitors in the distal lung, non-sorted epithelial cells were seeded in AOM +/−CHIR conditions. While strong SFTPC induction is initially detected by qPCR, extended

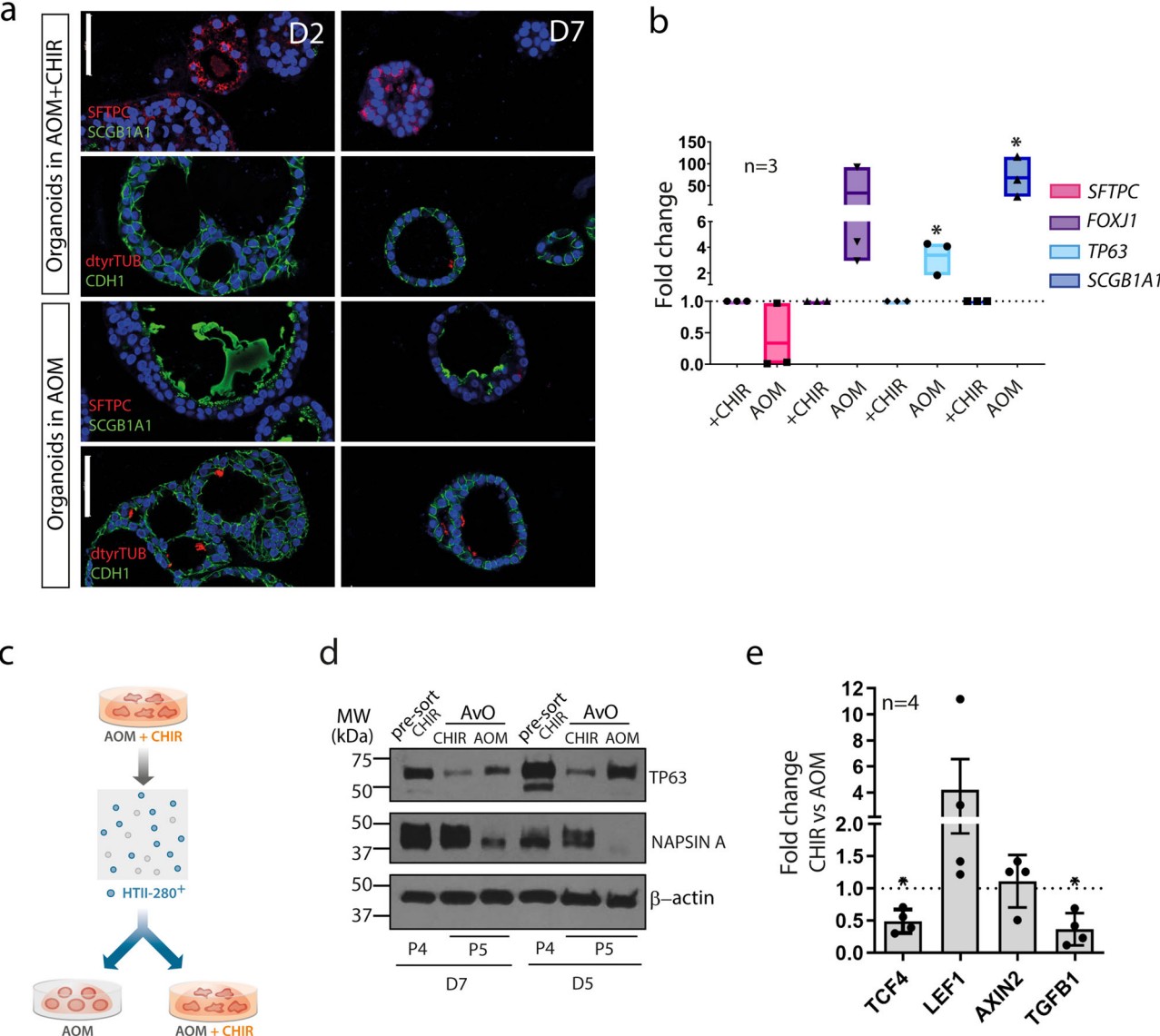

**Fig. 4 The differentiation route of HTII-280+ progenitors depends on the signaling environment. a** Comparative images from two different donor lines of HTII-280+-derived organoids grown in AOM reveal a shift in differentiation towards a higher expression of bronchial markers (SCGB1A1, dtyrTub), while SFTPC is only expressed in presence of the inhibitor (right image). Scale bar: 20 μm. **b** qPCR confirms a medium-dependent shift in phenotype. Data are represented as floating bars, min to max, with a line at mean. *n* = 3 independent donor lines.*$p = 0.038$ for *TP63* and *$p = 0.012$ for *SCGB1A1* calculated by Student´s *t*-test. **c** Layout of the resorting experiment. **d** Western blot of protein lysates showing medium-dependent regulation of the expression of bronchial (TP63) and alveolar marker (NAPSIN A) in newly formed organoids. Data represent two different alveolar donor lines in the stable expansion (P4) and corresponding resorted organoids. **e** Fold-change in mRNA expression level of Wnt target genes *TCF4*, *LEF1* and *AXIN2*, as well as *TGFB1* in HTII-280+-derived organoids under CHIR treatment. *n* = 4, where *n* represents independent donor lines. Data is presented as mean +/− SD *$p = 0.027$ for *TCF4*, and *$p = 0.049$ for TGFB1 calculated by two-tailed Student *t*-test.

cultivation led to a reversion to the baseline (Fig. 6a). These findings suggest that airway stem cells in the distal lung are resistant to "alveolar induction" by CHIR in contrast to HTII-280+ cells and that they have a competitive growth advantage. Indeed we found that early cultures of organoids in AOM medium (P0/P1) do have regions of HTII-280/SFTPC expressing cells, as illustrated by confocal imaging (Fig. 6b). However, SFTPC+ regions could not be detected in organoids ≥ P2. A rapid decrease in *SFTPC* expression during extended cultivation was confirmed by qPCR assay (Fig. 6c) and advanced organoid cultures (>2 months of expansion) expressed only markers of distal airway lineages.

Owing to the slower growth and smaller size even in the presence of CHIR, HTII-280+-derived alveolar organoids cannot be maintained in the long-term culture in parallel with airway progenitors, which explains the necessity to perform the step of early separation of HTII-280+ progenitors from the epithelial pool.

Next, we performed a comprehensive characterization of 7 airway lines from 5 donors (a total of 52,307 cells) by scRNA-seq of the long-term pool organoid cultures. Cluster annotation based on reference data from the Human Lung Cell Atlas[36] identified four broad classes of clusters: basal, secretory, ciliated, and proliferative (Fig. 6d, e and Supplementary Fig. 4a). A high degree of consistency in cell-type composition and proportions between different donors was observed for all samples, as well as the transcriptional similarity between the same cell type from different donors and CNV analysis again showed no notable

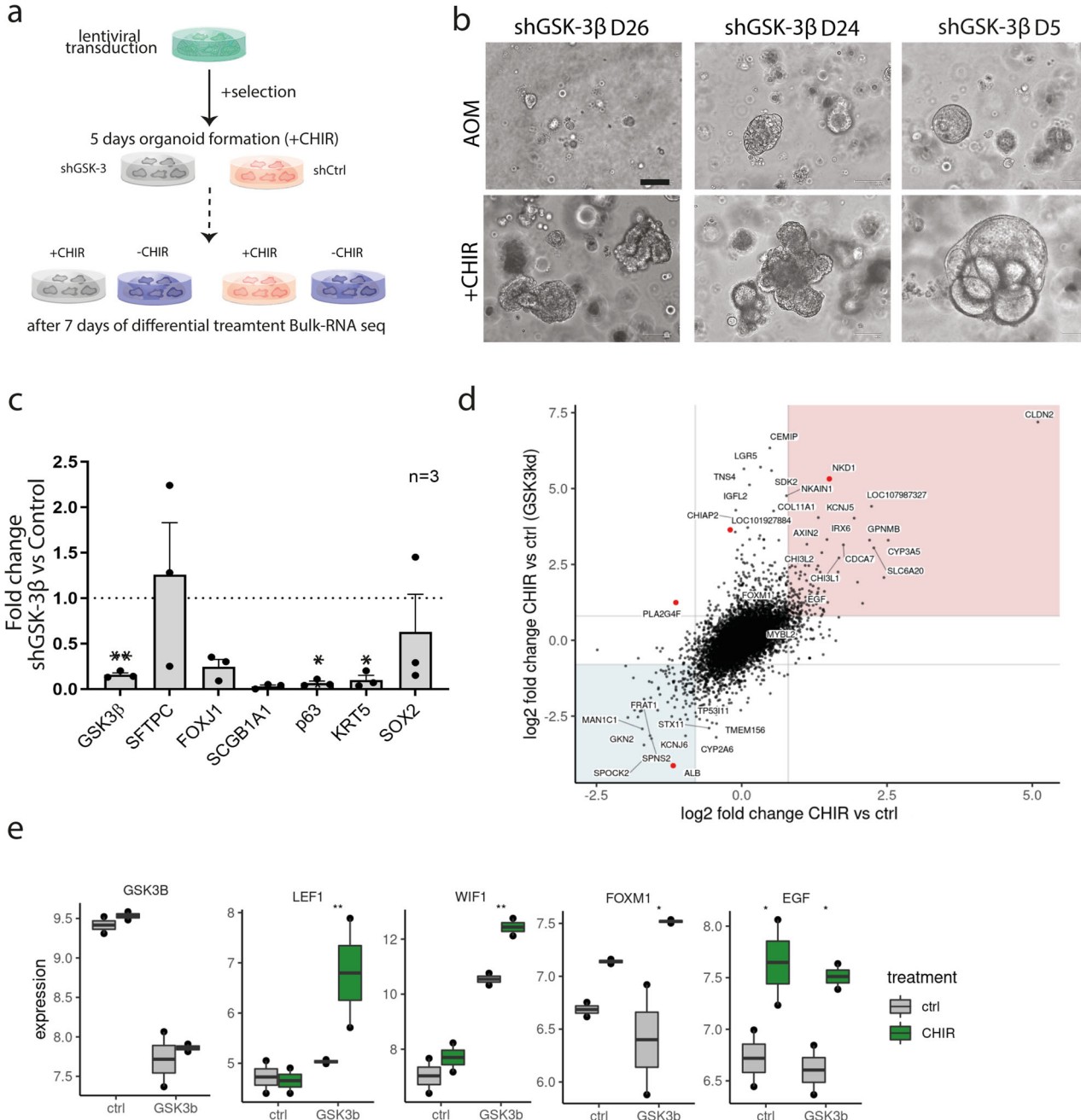

**Fig. 5 CHIR 99021 is required for alveolar organoids formation and differentiation. a** Experimental outline of GSK-3β knockdown in alveolar organoids. **b** shGSK-3β progenitors do not have organoid forming potential in medium without CHIR99021, as shown on representative images from three independent biological replicates. Scale bar: 100 μm. **c** Relative quantification of gene expression in shGSK-3β and control organoids reveals a strong reduction in expression of bronchial markers in presence of CHIR, while the SFTPC level remains not significantly altered. $n = 3$ of 3 independent pairs of donor lines (shControl and shGSK-3β). Data are presented as mean $+/-$ SD. **$p = 0.006$ for *GSK3β*, *$p = 0.011$ for *TP63*, *$p = 0.037$ for *KRT5*, calculated by two-tailed Students *t*-test. **d** Scatter plot of gene expression changes due to CHIR treatment in shGSK-3β and mock (non-mammalian shRNA) background derived from bulk RNA-seq data of two different donor organoid cultures. Genes with significant differences in the transcriptomic responses (adj. *p*-value < 0.05) are highlighted in red. Red and blue areas denote concordant responses with log2 fold-change > 0.8. **e** Normalized expression values of specific genes (GSK3B, Wnt targets LEF1 and WIF1, FOXM1, EGF) in both genetic backgrounds (ctrl and GSK3b) $+/-$ CHIR treatment (ctrl and CHIR). ***$p < 0.001$, **$p < 0.01$, *$p < 0.05$ (DESeq2 Wald test).

aberrations (Supplementary Fig. 4b, c). Importantly, our analysis revealed the existence of an expression gradient between basal progenitor populations (marked by *TP63/KRT5*) and secretory cell types (marked by *MUC5B/MUC5AC*) confirming the intrinsic property of organoids to recapitulate intermediate stages of cell differentiation in vitro (Supplementary Fig. 4d). This is complementary to confocal images revealing the presence of

domains of the multilayered epithelium and regions with an expanded basal layer (TP63), as well as differentiated epithelium, marked by terminally differentiated ciliated cells (Supplementary Fig. 4e) facing the lumen. Independent quantification of ciliated cells by imaging in three donors identified 2–4.5% ciliated cells similar to the range determined by scRNA-seq analysis (Supplementary Fig. 4f). While single-cell analysis could not

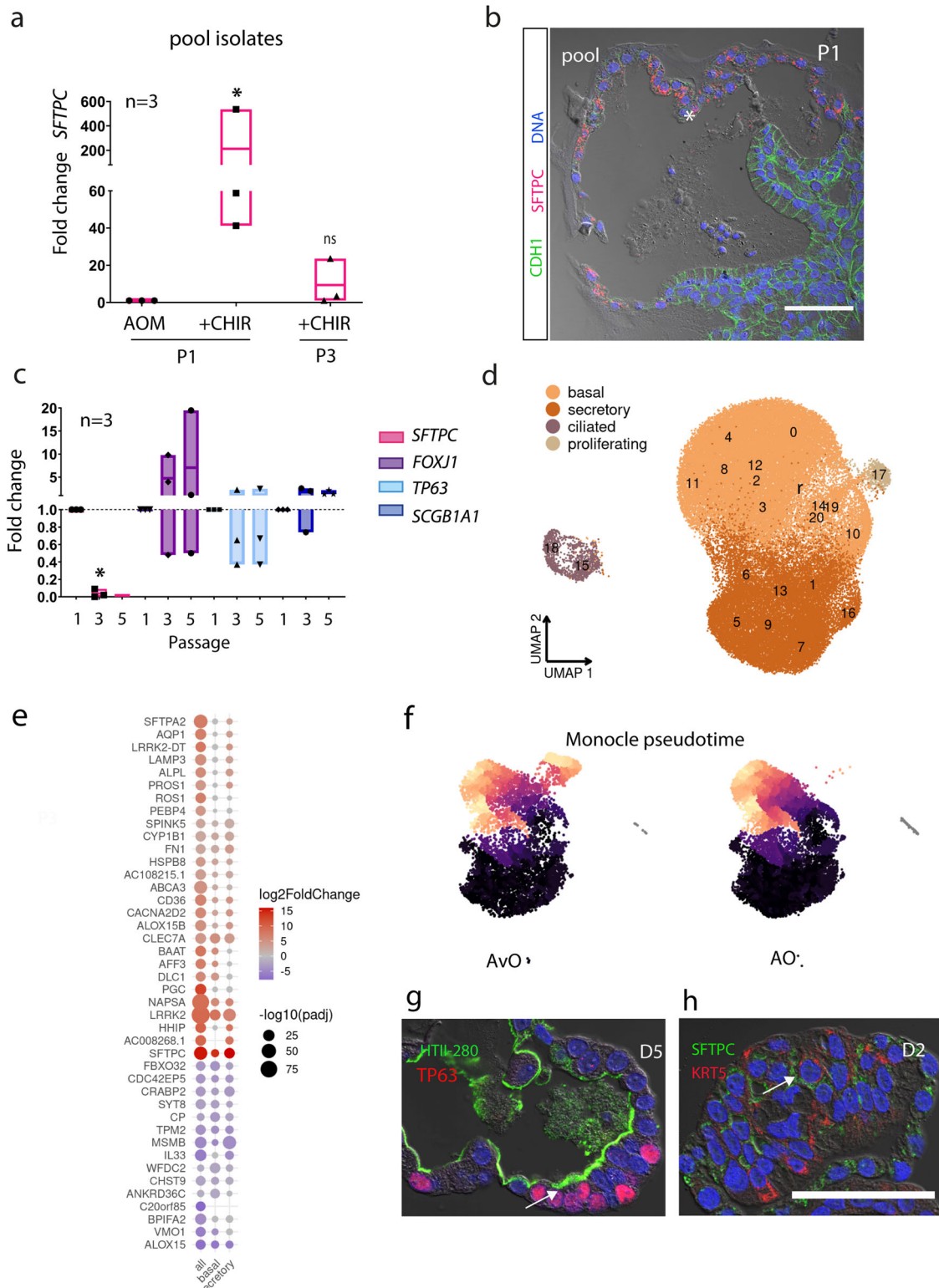

distinguish goblet and club cells within the "secretory" cluster, antibody staining against SCGB1A1 and MUC5AC demonstrated differentiation of club cells and goblet cells (Supplementary Fig. 4e). By testing variations in medium composition, we determined that in vitro preservation of stemness potentially depends on active NOTCH and SHH signaling as inhibition of those pathways by DAPT (10 μM) and Vismodegib (10 μM), respectively, prevents the organoid formation and long-term cultivation (Supplementary Fig. 5a). Similarly, the absence of

noggin and thus activation of the BMP pathway, as expected, led to an early growth arrest of two tested lines presumably due to loss of stemness (Supplementary Fig. 5b).

To directly compare the expression profiles of the alveolar and bronchial organoids from the same donor group, an analysis of the scRNA-seq was performed containing 3 pairs of respective organoid cultures. Overall cell types identified correspond well to the results of the individual group analysis (Supplementary Fig. 5c and Supplementary Data 2).

**Fig. 6 Airway progenitors cannot be primed by CHIR treatment to the alveolar lineage. a** Cultivation of organoids in the CHIR medium triggers a transient spike in *SFTPC* expression. $n = 3$ where $n$ are independent pool organoid cultures. Data are presented as floating bars, max to min, with a line at mean *$p = 0.028$ for condition AOM+CHIR, calculated by a two-tailed Student's $t$-test. **b** Immunofluorescence image of passage 1 (P1) of a lung organoid showing a large domain of SFTPC (asterisk, red) expressing cuboidal epithelium next to columnar airway epithelium (arrow, CDH1, green). Nuclei are counterstained with DAPI (blue) and tissue structure is visualized by differential interference contrast (DIC). Scale bar: 50 μm. **c** qPCR shows a drop in *SFTPC* expression during passaging (P1, P3, P5) of AO, while airway markers *FOXJ1*, *TP63*, and *SCGB1A1* remain stable. $n = 3$, where $n$ are independent donor lines. *$p = 0.043$ for *SFTPC*, by two-tailed Student's $t$-test. **d** Unsupervised clustering results (numbers) and broad classification (colors) in lung organoid scRNA-seq data ($n = 5$ donors, 63,826 cells). **e** Direct comparison of gene expression between HTII-280+-derived organoids and pool airway organoids showed upregulation of alveolar hallmark genes in all cell types. **f** Monocle 3 "pseudotime" analysis of HTII-280+-derived alveolar organoids and pool airway organoids showing differentiation path (dark dots representing progenitors and brighter colors more differentiated cells). **g** Immunofluorescence stainings of alveolar (AvO) organoids showing HTII-280+ cells co-expressing airway marker TP63 (arrow) and in **h** SFTPC+/KRT5+ cells (arrow). Confocal images are representative of staining from three different donor lines of AvO organoids. Nuclei are counterstained with DAPI, and tissue is visualized by DIC. Scale bar: 50 μm.

Differential expression analysis between cells from HTII-280+-CHIR organoids and the corresponding clusters in pool organoids from the same donors revealed a systemic shift and abundant expression of surfactant proteins not only in the designated AT2 cluster but also in basal and secretory cells (Fig. 6e and Supplementary Data 2) again illustrating phenotypic plasticity of organoid cells derived from HTII-280+ progenitors. Analysis of differentiation trajectories of cell types in both types of organoids by Monocle 3 "pseudotime" (Fig. 6f) showed that HTII-280+-derived alveolar organoids do contain progenitors (dark blue) and a continuous set of intermediary expression profiles including differentiated AT2 cells (yellow) while the differentiation program of airway organoids includes only bronchial cell types. Indeed immunofluorescence staining of alveolar organoids with markers of the airway and alveolar differentiation confirmed the existence of intermediate cell types co-expressing HTII-280 and TP63 as well as SFTPC and KRT5 proteins (Fig. 6g, h). Taken together, our data consistently support the model of a complex in vitro differentiation mechanism of HTII-280+ progenitors without strict separation between bronchial and alveolar cell types suggesting a high degree of plasticity.

**Influenza infection organoid model recapitulates tissue response and preserves epithelial integrity.** After generating stable organoid cultures representing different epithelial compartments of the lung, we aimed to establish corresponding infection models with human influenza virus (IAV) strain H3N2. Pool airway organoids and HTII-280+-CHIR alveolar organoids from the same donor were infected in parallel. Immuno-fluorescence staining confirmed the presence of the virus in both AT2 and bronchial cell types (Fig. 7a). The productivity of the viral replication in organoids was confirmed by plaque assay, which revealed a sustained level of infectious particles retrieved over 5 days irrespective of donors (Fig. 7b). IAV readily infects organoids and achieves a titer of ~$10^6$ PFU/ml at 16 h post-infection (hpi), a one-log rate higher than in the ex vivo IAV infection model of human lung tissue explants (Supplementary Fig. 6a). The slight increase in viral load recovered from HTII-280+-CHIR organoids in comparison to airway organoids was determined to be non-significant. Despite considerable stress response apparent on phase-contrast images of the infected cultures, organoids do survive the acute infection and preserve epithelial integrity, while infected cells are shed into the lumen (Supplementary Fig. 6b). Single-cell RNA-seq analysis of IAV-infected organoids confirmed an exceptionally high infection rate, as viral transcripts were detected in virtually all cells. Based on the co-expression of 4 key viral genes (PA, PB1, PB2, and NP)[37] within a host cell, we distinguished likely productive from likely non-productive infection, confirming a higher infection rate

compared to the ex vivo lung tissue (Supplementary Fig. 6c). The high viral load impacted cluster analysis, as a large group of cells was classified as "infected" with an underlying cell identity not distinguishable (Fig. 7c). Accordingly, a strong inflammatory response was detected in all cell types (Fig. 7c, right panel and Supplementary Data 2). Robust induction of interferon signaling was detected in both alveolar and bronchial organoids, as shown by the scatter plot of gene expression changes (Fig. 7d). We asked how the in vitro organoid IAV model relates to the infection response of the epithelial compartment in lung tissue explants[38]. Singe-cell RNA-seq of two donors, infected ex vivo for 16 h, confirmed IAV infection in different cell types (immune, epithelial, endothelial; Fig. 7e and Supplementary Fig. 6d). Although the infection rate in the lung epithelial compartment was considerably lower than in organoids (Supplementary Fig. 6c), differential gene expression analysis showed a comparable transcriptomic response of tissue epithelium and the organoid model (Fig. 7f and Supplementary Data 2).

## Discussion

Our results show that a lineage-committed differentiated subpopulation of AT2 cells, expressing surface antigen HTII-280, has the stemness capacity to replenish the alveolar lineage but also can give rise to bronchial cell types with basal, secretory, and ciliated phenotypes. By scRNA-seq transcriptomic analysis of HTII-280+/EpCAM+ cells, we showed that the compartment contains subpopulations of cells with distinct expression profiles and points to a small group of cells that exhibits similarity with adult stem cells in other epithelial tissues. Confocal imaging of epithelial cell isolates, as well as imaging of sections from native lung tissue, confirmed the expression of the *FOXM1* transcription factor, an important regulator of cell cycle transition phases (G1 to S and G2 to M)[39], in a subset of AT2 cells. Also, in the same subcluster SCENIC analysis identified enrichment of *MYBL2* regulated transcriptional network. More studies are needed to determine the exact localization and mechanisms of regulation of these progenitors in the alveolar region of the adult lung, but in vitro growth properties of the organoids suggest critical involvement of the WNT, NOTCH, BMP, and FGF pathways. Moreover, upregulation of *FOXM1* in organoids in CHIR99021 medium, independently of GSK-3β levels, supports the hypothesis that this gene could be functionally important for the induction of the alveolar phenotype. Notably, its expression appears not directly mediated by Wnt signaling, though it can not be excluded that crosstalk occurs during alveolar differentiation.

Comparison of the single-cell expression profiles from freshly isolated HTII-280+ cells with resulting cell types from the organoids they generate reveals the whole spectrum of intermediate phenotypes that are suggestive of dynamic reprogramming in vitro. This property of HTII-280+ cells to enter diverging

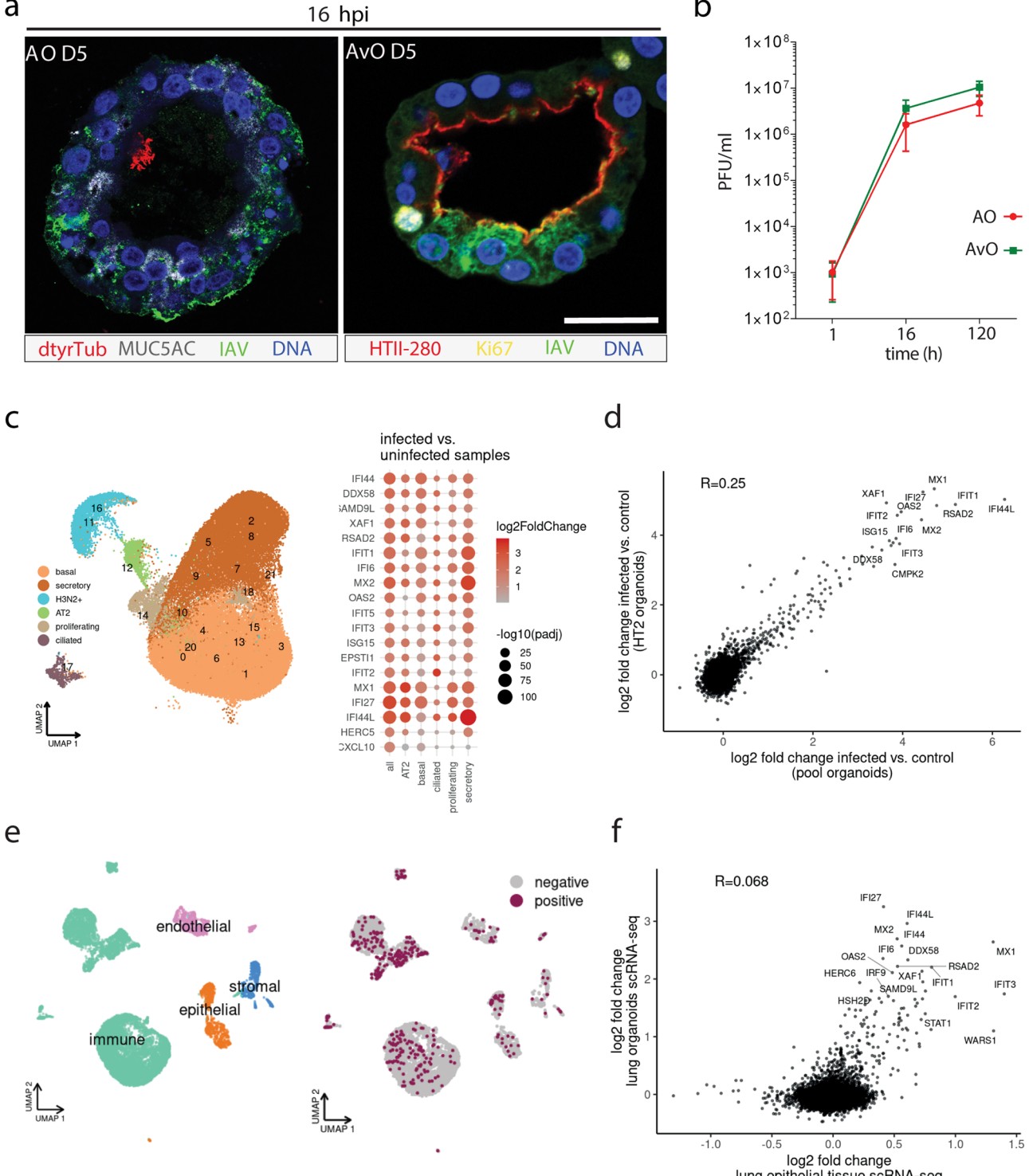

**Fig. 7 IAV infection of the organoids mimics lung tissue response. a** Confocal images of IAV-infected airway organoids (AO) (IAV, green, MUC5AC, gray, dtyrTub, red), and alveolar organoids (AvO) (HTII-280, red, Ki67, yellow). Scale bar: 20 μm. **b** Virus titers of the organoid infection in paired organoid lines over a 5-day time course reveal robust and sustained replication. Data represents $n = 3$ of paired organoid lines from different donors. Error bars ($+/-$ SEM) are calculated from the number of plaque-forming units in the MDCK monolayer. The difference in replication between organoid types was determined to be non-significant. **c** scRNA-seq reveals IAV-dependent shift in cell type classification and appearance of an infected cluster ($n = 3$ donors, 53090 cells) (left). Differential gene expression analysis between infected samples and non-infected controls reveals a broad interferon response across all cell types (right). **d** Correlation of the antiviral response between bronchial and alveolar organoids. **e** Single-cell sequencing results of identified cell types from human lung tissue explants and distribution of IAV infection ($n = 2$ donors, 12368 cells). **f** Correlation of antiviral response between organoids and lung tissue epithelium.

differentiation routes in vitro has not been identified in previous studies, which reported long-term expansion of alveolar organoids. This could potentially be due to differences in the medium components that we used. A detailed comparison of cultivation conditions revealed that the presence of recombinant EGF and the absence of MAP kinase inhibitor in other studies could contribute to the differences in phenotype[9,10]. Indeed we could show that recombinant EGF leads to further upregulation of SFTPC in AT2 cells. Also, the only study that reported successful maintenance of alveolar organoids in vitro without CHIR99021 used recombinant Wnt agonists for Wnt activation (Wnt3a and RSPO1) but also a range of additional growth factors among which are neuregulin-1, IGF1, HGF, SDF1, IL1-6, which essentially confirms our findings that CHIR99021 is considerably more pleiotropic than previously described[40]. Notably, bulk RNA-sequencing of alveolar organoids also demonstrated that CHIR treatment promotes the expression of endogenous EGF, which also suggests the importance of the pathway for alveolar differentiation. A recent study identified the potential of AT2 cells to transdifferentiate to KRT5+ cells[21] and investigated the role of lung mesenchymal cells in this process. They define a new phenotype of alveolar basal intermediate cells (ABI)[21] that closely resembles intermediate phenotypes we have found in alveolar organoids grown in CHIR99021 medium (Figs. 3 and 6g). However, the study focused on the complex co-culture model with mesenchymal cells and is limited to analysis of the short-term organoid culture (21 days), which is a notable distinction from our model that is based on stable maintenance of stemness potential in vitro (cultivation >6 months). We clearly show that the inherent plasticity of HTII-280+ cells is a stable biological property, continuously present and the differentiation route can be modified by subtle changes in the paracrine signaling environment. Subsorting of HTII-280+ progenitor cells from alveolar organoids expanded for >3 months (Fig. 4c and d) followed by seeding in medium with and without CHIR demonstrated that plasticity of transdifferentiation potential is preserved. Notably, alveolar organoids in our long-term culture showed a relatively high level of TP63 protein at the time of subsorting, in agreement with scRNA-seq data, but the resulting new organoid lines showed more clearly diverging alveolar and airway phenotypes. This suggests that bronchial properties of organoids containing intermediate cell types can be reversed, which could have important clinical implications, and warrants future detailed research of this phenomenon. While our study provides ample evidence that airway progenitors do outcompete HTII-280+ cells in the primary culture even in growth conditions favoring alveolar differentiation, further lineage tracing experiments are needed to better characterize this phenomenon.

Taken together, we demonstrate that a single inhibitor, CHIR99021, alters the lineage commitment of HTII-280+ progenitors and is required to induce alveolar differentiation. Previous studies using CHIR99021 focused solely on its role as a Wnt pathway activator[33] and suggest significant importance of a localized amplification of the Wnt pathway signal. Indeed, distinct populations of mesenchymal cells expressing LGR5 and LGR6 receptors have been identified as mediators of niche-specific Wnt signals in the distal airways and alveoli in the mouse[41] and similar mechanisms could be responsible for controlling epithelial differentiation in the human lung. Nevertheless, it is notable that we could not detect a significant increase in expression of Wnt target genes in HTII-280+-CHIR organoids when compared to HTII-280+-AOM, which show a clear airway phenotype. To further decipher the effect of CHIR and the role of Wnt signaling, we performed shRNA-mediated knockdown of the main CHIR target and negative Wnt pathway regulator GSK-3β in alveolar organoids followed by bulk RNA-sequencing of +/−CHIR-treated organoids.

Interestingly, silencing of GSK-3β abrogated organoid forming capacity of alveolar progenitors in a medium without CHIR99021, a finding that proves that action of this inhibitor extends beyond GSK-3β. Moreover, while the expression profiles of +/−CHIR-treated organoids remained similar, there were many genes regulated solely by the presence of CHIR and not GSK-3β. Among these genes, we identified FOXM1 and EGF, which further substantiates the potential involvement of these two proteins in the regulation of alveolar stemness and differentiation. Overall, our mechanistic data by genetic manipulation convincingly demonstrate that the CHIR-mediated effect on AT2 cell differentiation appears to be driven by additional signaling pathways and cannot be explained by targeted activation of the Wnt pathway. More extensive studies and validations are needed to investigate the pleiotropic role of CHIR, as well as the involvement of EGF and MAP kinase pathways in the control of bipotency and stemness potential in the alveolar compartment. While the generation of stable organoid lines from HTII-280+ sorted cells is a robust methodology, we did observe some variation in the frequencies of fully differentiated AT2 cells, potentially reflecting differences in genetic background or conditions in the local tissue environment at the time of sample collection. More studies are needed to identify the exact molecular mechanisms that control lineage commitment of the epithelial progenitors in response to exogenous signaling cues, as well as identify cell types that regulate the niche environment in vivo.

A systematic comparison of HTII-280+ stem cell properties in the tissue and organoid lines they generate from different patients (e.g., COPD, IPF) could prove to be of pivotal importance for identifying early events and cellular mechanisms of chronic lung diseases. As demonstrated here, the organoid model showed comparable results to primary human lung explant tissue with regard to cellular composition and response to IAV infection, which demonstrates their suitability for studying infectious diseases. Organoid biobanks generated from HTII-280+ cells representing bronchial and alveolar parts of the respiratory tract of a particular human donor could be a rapidly available resource to test for emerging infections such as IAV or coronaviruses. Moreover, they could complement the results of animal experiments by avoiding species particularities, especially in the investigation of zoonotic diseases.

## Methods

**Sample acquisition.** Peripheral lung tumor-free explants were obtained from lung cancer patients who underwent lung resection surgery. The study was approved by the ethics committee at the Charité clinic (project EA2/079/13). Written informed consent was obtained from all patients.

**Cell lines.** 293T HA Rspo1-Fc cell line, R&D, 3710-001-01, MDCKII cell line (CRL-2936).

**Organoid cultivation.** After thorough washing with HBSS, the lung tissue, tissue was minced with scissors and transferred to an enzyme mixture containing 500 U/ml Collagenase I (Gibco), 5 U/ml Dispase II (Gibco), and 1 U/ml DNase (Applichem) in HBSS supplemented with 10 μM Y-27632 dihydrochloride (Tocris)[42]. The minced tissue was incubated for 45–60 min in a shaking water bath at 37 °C before being vigorously vortexed and passed through a sieve to remove undigested tissue pieces. Cells were centrifuged (300 × g, 5 min), resuspended in red blood cell (RBC) lysis buffer (Invitrogen), and incubated for 5 min at RT to remove erythrocytes. Subsequently, cells were washed with ADF++ (Advanced DMEM/F12 (Invitrogen) with 10 mM HEPES (Invitrogen) and 1x GlutaMax (Invitrogen)), counted, and seeded in the extracellular matrix substitute Cultrex (R&D) at a concentration of ~2000 cells/μl. Upon solidification of the gelatinous matrix, the following medium, adapted from[8], was added on top of the cultures to induce the growth of lung organoids: ADF++ supplemented with 10% R-spondin1 conditioned medium (produced as described previously[43] using the 293T HA Rspo1-Fc cell line), 1x B27 supplement (Invitrogen), 1x Primocin antibiotic mix (Invivogen), 1.25 mM N-Acetylcysteine (Sigma), 5 mM Nicotinamide (Sigma), 0.5 μM SB202190 (Sigma), 1 μM A83-01 (Merck), 100 ng/μl human Noggin (Peprotech), 100 ng/μl human FGF10 (Peprotech) and 25 ng/ml human FGF7 (Peprotech). Y-27632 dihydrochloride (10 μM, Tocris) was added only in the first week of

culture[8]. The GSK3 inhibitor CHIR99021 (Sigma, 3 µM), as well as the SHH inhibitor Vismodegib (Selleckchem, 10 µM) and the NOTCH inhibitor DAPT (Sigma, 10 µM) were added to the cultures as indicated. Organoids were kept in an incubator at 37 °C, 5% $CO_2$.

Upon growth, for 2–4 weeks organoids were expanded by enzymatic digestion. To this end, organoids were released from Cultrex with cold ADF++, centrifuged (300 x g, 5 min), incubated in TrypLE Express Enzyme (Gibco) for ~5–6 min at 37 °C, and briefly vortexed. After washing with ADF++, cells were resuspended in Cultrex at a ratio of 1:3 to 1:8.

**Virus strain**. Human seasonal influenza virus A/Panama/2007/1999 (Pan/99[H3N2]) (short IAV) was propagated on MDCKII cells. Viral stocks were aliquoted, stored at −80 °C, and titrated on MDCKII cells by plaque assay.

**Fluorescence-activated cell sorting (FACS)**. For the isolation of HTII-280+ cells, lung tissue was digested as described above. After removing undigested tissue residues with a sieve, the suspension was successively filtered through 100, 70, and 40 µm cell strainers. Filtered cells were centrifuged and erythrocytes removed by the addition of RBC lysis buffer. Washing with ADF++ was followed by resuspension in 300 µl staining buffer (ADF++ with 1x N2 (Invitrogen), 1x B27, 5 mM Nicotinamide, 1.25 mM N-Acetylcysteine, 10 µM Y-27632 and 5% FCS (Capricorn Scientific). Mouse anti-HTII-280 IgM antibody (Terrace Biotech, TB-27AHT2-280) was added at a dilution of 1:50 and incubated for 30 min on ice. Cells were washed with 2 ml ADF++ supplemented with 5% FCS, resuspended again in 500 µl staining buffer, and incubated for 30 min on ice with the secondary goat anti-mouse IgG (H+L) A488 antibody (Thermo Fisher, A-11017) at a dilution of 1:500. Control without primary but secondary antibody staining was kept in parallel. Upon incubation, cells were washed with 2 ml ADF++ with 5% FCS and resuspended in DPBS (Gibco) supplemented with 10 µM Y-27632 dihydrochloride and 1x B27, before being passed through a 40 µm strainer into FACS tubes. Finally, the 7-AAD viability staining solution (Biolegend, 420403) was added 10 min before the sorting at a dilution of 1:100. Cell sorting was performed by the FACS Core facility using a BD FACSAria™ II cell sorter (BD Biosciences). Sorted cells were seeded in Cultrex at a concentration of ~1000 cells/µl and overlaid with the respective organoid medium (see above) upon solidification. For scRNA-seq, immediately after sorting $0.5–1 \times 10^6$ HTII-280+ cells were fixed with methanol as described above and stored at −20 °C.

**Cytospin**. To identify progenitor subpopulations of HTII-280+ cells, $2–5 \times 10^4$ cells of HTII-280+ and HTII-280+/EpCAM+ were placed in cytospin cones and spun at 500 rpm for 3 min on a Shandon Cytospin 3 (Thermo Fisher Scientific) on a glass slide. Subsequently, slides were air-dried at room temperature for 5 min and then fixed with 4% PFA for 5 min. Afterwards, the slides were washed with three changes of PBS for a total of 15 min on a rocker and permeabilized for 15 min with 1% Triton in PBS. The regular staining protocol for immunofluorescence (described below) was followed, with the FOXM1 (1:50, Cell Signaling, 5436 and HTII-280 1:200 Terrace Biotech, TB-27AHT2-280) primary antibodies diluted in IF staining buffer and incubated overnight at 4 °C:

**Western blot**. Organoids were washed with cold PBS, pelleted, lysed in 1x Laemmli buffer (Bio-Rad), and heated for 10 min at 95 °C. Lysates were subjected to western blot as described previously[44]. Briefly, protein extracts were separated on a 4–15% or 4–20% Mini-PROTEAN® TGX™ Precast Protein Gel (Bio-Rad) and transferred to a PVDF membrane (Merck Millipore). Membranes were blocked with Odyssey blocking buffer (LI-COR Inc.) and probed with antibodies against HOPX (1:500, Proteintech, 11419-1-AP), HTII-280 (1:250, Terrace, TB-27AHT2-280), NAPSIN A (1:500, Cell signaling, 62434), p63 (1:1000, Abcam, ab124762), SCGB1A1 (1:500, R&D, MAB4218), SFTPC (1:500, Millipore, AB3786), and actin (1:5000, Sigma, A1978). All antibodies were diluted in Odyssey blocking buffer and incubated ON, 4 °C.

Proteins were detected by incubation with HRP-conjugated IgG antibodies HRP anti-mouse (Pierce 32230) 1:10,000 HRP goat anti-rabbit (sc-2004) 1:5000, HRP goat anti-rat (ab97057) 1:10,000 and Amersham ECL Prime Western Blotting System (Cytiva) or Pierce ECL Western Blotting Substrate (Thermo Fisher Scientific). The membranes from each experiment were exposed using Carestream Biomax Light Film (Sigma Aldrich).

**Immunostaining and microscopy**. Organoids were fixed with 4% PFA for 1 h at RT and washed two times with PBS. Subsequently, organoids were embedded in HistoGel™ (Thermo Fisher) and subjected to paraffin embedding and slicing. Sections of organoids were routinely checked with HE staining. Immunofluorescence (IF) stainings were conducted for visualization of detailed organoid phenotypes and differentiation status. Briefly, organoid sections were deparaffinized with Roticlear® (Carl Roth) and rehydrated with decreasing ethanol concentrations (100%, 96%, 80%, 70%, 50%). After washing with 0.01 M PBS, Antigen retrieval was performed by heating in TRIS/EDTA (30 min, 100 °C). Next, slides were washed and permeabilized for 15 min with 1% Triton in 0.01 M PBS. Washing with PBS was followed by blocking (30 min, RT) with either 5% goat or 5% donkey serum in IF staining buffer (0.01 M PBS with 1% BSA, 0.05% Tween-

20). The following primary antibodies were diluted in IF staining buffer and incubated overnight, 4 °C: acetylated Tubulin (1:100, Sigma, T7451), detyrosinated Tubulin (1:100, Abcam, ab48389), E-Cadherin (1:100, BD Biosciences, 610181), HTII-280 (1:200, Terrace, TB-27AHT2-280), FOXM1 (1:100, Abcam, ab20729) Influenza A (1:100, Serotec, OBT1551), Keratin 5 (1:500, Biolegend, 905501), Ki67 (1:400, Cell Signaling, 9027S), MUC5AC (1:400, Merck, MAB2011), p63 (1:1200, Abcam, ab124762), SCGB1A1 (1:200, R&D, MAB4218), SFTPC (1:1000, Millipore, AB3786). Washing was followed by incubation with the corresponding secondary antibodies goat-anti-mouse IgG488 (A-11017), goat anti-rat IgG Alexa 488 (A-11006), goat anti-rabbit IgG Alexa 594 (A-11072), goat-anti-mouse IgG Alexa 555 (A-32727), donkey-anti-rabbit IgG Alexa 546 (A10040), donkey-anti-goat IgG Alexa 488 (A11055), donkey anti-mouse IgG Alexa594 (A-21203), chicken anti Rat488 (A-21470) (Thermofisher Scientific), diluted 1:2000 in IF staining buffer. Nuclei were subsequently counterstained with DAPI (Sigma Aldrich).

Immunofluorescence was analyzed by spectral confocal microscopy using an LSM 780 [(objectives: Plan Apochromat 40x/1.40 oil DIC M27 and Plan Apochromat 63x/1.40 oil DIC M27), Carl-Zeiss, Jena, Germany]. To reveal organoid and cell morphology, images were combined with Differential Interference Contrast (DIC). Images were processed using ZEN 2012.

**Transmission electron microscopy**. The organoids were fixed with 1.5% PFA and 1.5% glutaraldehyde (both from Serva, Heidelberg, Germany) in 0.15 HEPES buffer. After post-fixation in 1% $OsO_4$ (Electron Microscopy Sciences, Hatfield, USA) in 0.1 M cacodylate buffer at RT for 2 h, the samples were embedded in agarose for 1.5 h at 4 °C, followed by incubation in half-saturated aqueous uranyl acetate (Merck, Burlington, USA) ON at 4 °C. After dehydration in a graded acetone series, the samples were transferred to epon resin (Roth, Karlsruhe, Germany). Finally, sliced ultrathin sections of 70 nm thickness were stained with uranyl acetate and lead citrate. Samples were examined using Zeiss EM 906 at 80 kV acceleration voltage (Carl Zeiss, Oberkochen, Germany).

**qPCR**. For total RNA isolation, the RNeasy Mini Kit (Qiagen) was used according to the manufacturer`s instructions. Organoids were released from Cultrex with cold PBS, pelleted, and resuspended by vortexing in 350 µl RLT lysis buffer supplemented with β-mercaptoethanol. Upon RNA isolation, 0.3 µg RNA was reverse transcribed using a high-capacity cDNA reverse transcription kit (Thermo Fisher). Subsequently, quantitative PCR was performed using TaqMan assays on an ABI 7300 instrument (Thermo Fisher). TaqMan™ Master-Mix (Thermo Fisher) was used in combination with the following TaqMan probes (Thermo Fisher): FOXJ1 (Hs00230964_m1), SCGB1A1 (Hs00171092_m1), SFTPC (Hs00951326_g1), TP63 (Hs00978340_m1), HOPX (Hs05028646_s1), LEF1 (Hs01547250_m1), TCF4 (Hs00162613_m1), AXIN2 (Hs00610344_m1), TGFB1 (Hs00998133_m1), GSK3B (Hs01047719_m1), KRT5 (Hs00361185_m1), SOX2 (Hs01053049_s1), FOXM1 (Hs01073586_m1), and GAPDH (Hs02758991_g1). Relative gene expression was calculated by average ΔCt and ΔΔCt values using glyceraldehyde-3-phosphate dehydrogenase (GAPDH) as a housekeeping gene for normalization. Statistical analysis was performed with the GraphPad Prism 6 software. Differences in mRNA expression levels were calculated using a two-tailed student's t-test on ΔCt values.

**Infection of organoids**. Mature human lung organoids were collected on ice to remove the remaining matrix and broken up by repeated resuspension using a disposable syringe with a 27G needle. Before infection one organoid well was dissociated into single cells (as described above) to determine the cell number. Organoid fragments were then either mock-infected with infection medium (ADF++) or challenged with influenza A virus (MOI 0.4) for 1 h at RT. After infection, organoids were washed with ADF++, resuspended in Cultrex, and overlaid with the respective organoid medium (as described above). Samples for scRNA-seq and IAV replication were taken at indicated time points. To ensure retrieval of all infectious particles generated within a given time in the culture, the gelatinous matrix was dissolved by pipetting up and down with the overlaid medium. Then the mixture of medium, matrix, and organoids was incubated for 5 min on ice and centrifuged (300 × g, 5 min). The supernatant was transferred to new tubes and frozen at –80 °C.

**Lentiviral transduction of Organoids**. Lentiviral particles with two different shRNA sequences against GSK-3β (shRNA_1: CATGAAAGTTAGCAGAGACAA, TRCN0000039564; shRNA_2: CCCAAACTACACAGAATTTAA, TRCN0000039999), as well as control transduction particles with a non-mammalian shRNA (SHC002V) were purchased from Sigma.

For transduction of the shRNAs HTII-280+-derived CHIR-treated organoids were disrupted by enzymatic (TrypLE, 15 min, 37 °C) and mechanical (3 times through 27G needle) treatment. Single cells were washed and counted. In all, 200,000 cells were resuspended in 250 µl organoid medium supplemented with ROCK inhibitor and Polybrene (8 µg/ml) and mixed with the amount of virus to achieve MOI 1. The suspension was pipetted into wells coated with a 1:1 mixture of Cultrex and ADF++. The suspension was incubated for 16 h at 37 °C. The next day the cells were taken up with 1 ml ADF++ and centrifuged. The cells were seeded into Cultrex and overlaid with organoid medium supplemented with ROCK inhibitor. Two days after seeding cells were selected with Puromycin (1 µg/ml) for

1 week. Two different shRNAs against GSK3b were transduced and one control non-mammalian shRNA.

**Bulk RNA sequencing**. RNA was isolated from the organoids using Qiagen RNAeasy kit following to the manufacturer's protocol. Sequencing libraries were prepared using the NEBNext Ultra II Directional RNA Library Prep Kit for Illumina (New England Biolabs) and sequenced on a NextSeq 500 device (Illumina) using a 75 cycles high output kit.

Reads were mapped using STAR (version 2.7.3a) against the GRCh38 genome and gene expression was quantified with featureCounts (version 2.0.0) using the Gencode v33 reference. Differential expression analysis was performed in R (version 4.0.3) with DESeq2[45] (version 1.30.1).

**Single-cell preparation from organoids**. Single cells were generated from organoids by enzymatic digestion with the TrypLE Express enzyme (12–15 min, 37 °C), followed by mechanical disruption through vigorous vortexing. Subsequently, cells were washed with ADF++, passed multiple times through a 27G needle, and filtered with a 40 μm strainer to remove any remaining cell clumps. To collect several samples for simultaneous scRNA-seq, cells were exposed to methanol fixation. To this end, cells were washed twice by centrifugation (300 x *g*, 5 min) with 1 ml cold DPBS before being resuspended in 200 μl cold DPBS. Next, 800 μl ice-cold methanol was added dropwise to the cell suspension. The cells were then stored at –20 °C and rehydrated shortly before scRNA-seq by equilibration to 4 °C, centrifugation ($1000 \times g$, 5 min), and resuspension in rehydration buffer (0.04% BSA, 1 mM DTT, 0.2 U/μl RNase inhibitor in 3x SSC buffer (Sigma).

Before performing scRNA-seq, fixed as well as fresh cells were diluted to a concentration of 10,000 cells/μl in rehydration buffer or ADF++, respectively.

**Infection of lung tissue and single-cell preparation**. Immediately after arrival tumor-free peripheral lung tissue explants were cut into small pieces (0.5 x 0.5 x 0.5 mm, 100–200 mg) and incubated overnight in RPMI 1640 medium (Merck Biochrom) at 37 °C, 5% $CO_2$ to wash off clinically applied antibiotics. Lung infection experiments were done in RPMI 1640 medium supplemented with 0.3% bovine serum albumin and 2 mM L-glutamine at 37 °C, 5% $CO_2$ as described[38,46]. For infection tissue pieces were inoculated with either 200 μl control medium or 200 μl of $1 \times 10^6$ plaque-forming units (PFU) IAV diluted in control medium per 100 mg tissue. The infection and control medium were injected using a disposable syringe with a needle (27G) to assure proper stimulation. After incubation for 1 h at 37 °C, the excess virus was removed by two washing steps with PBS. The medium was added before incubation at 37 °C, 5% $CO_2$. Samples were taken at the indicated time points. For single-cell, preparation lung tissue was minced and placed in a digestion medium (500 U/ml Collagenase, I.5 U/ml Dispase, and 1 U/ml DNAse) for 1 h at 37 °C. Cells were then filtered through a 70 μm strainer and the enzymatic reaction was stopped by adding cold RPMI with 10% FCS and 1% L-Glutamine. Cells were washed with cold RPMI and red blood cells were lysed using RBC lysis solution (MiltenyiBiotec). Finally, cells were filtered using a 40 μm Flowmi® Cell Strainer (Millipore) and resuspended in PBS supplemented with 2% FCS at a concentration of 10,000 cells/μl for scRNA-seq.

**Infectious particle quantification**. IAV infectious particles were quantified by plaque titration on MDCKII cells. Briefly, MDCKII cells were seeded in 12-well plates to form a monolayer and incubated with virus-containing cell culture supernatants at different dilutions for 45 min. Subsequently infected cultures were overlaid with a 1:1 mixture of 2.5% Avicel medium and 2xMEM (Gibco) supplemented with 0.2% BSA (PAA), 0.001% Dextran, 0.05% $NaHCO_3$ and 1 μg/ml TPCK-treated Trypsin (Sigma). After 48 h, cells were washed twice with PBS and plaques were fixed and visualized by staining with crystal violet.

**Single-cell RNA sequencing (scRNA-seq)**. For organoid as well as lung tissue samples single-cell capturing and downstream library constructions were performed using the Chromium Single Cell 3′ V3.1 library preparation kit according to the manufacturers' instructions (10x Genomics). Full-length cDNA tagged with cell-barcode identifiers was PCR-amplified and sequencing libraries were prepared. The constructed libraries were either sequenced on the Nextseq 500 using 28 cycles for read 1, 55 cycles for read 2, and 8 index cycles, or on the Novaseq 6000 S1 using 28 cycles for read 1, 64 cycles for read 2, and 8 index cycles, to a median depth of 36,000 reads per cell.

**Analysis of scRNA-seq data**. The Cell Ranger Software Suite (Version 3.1.0) was used to process raw sequencing data with the GRCh38 reference. For infected samples, we used GRCh38 augmented by the IAV (DQ487333.1) genome. We then used CellBender (https://doi.org/10.1101/791699) to remove background RNA. Single-cell RNA-sequencing data analysis was performed in R (version 4.0.1) with Seurat Cells with at least 500 and less than 5000 detected genes, less than 25,000 UMIs, and less than 10% mitochondrial content were combined from each library, library depth (total number of UMIs), and cell cycle

scores were regressed out when scaling data. Libraries from different donors were then integrated using "Integrate Data", followed by again regressing out library size, mitochondrial content, and cell cycle scores followed by doublet removal with DoubletFinder (version 2.0.3). For the HTII-280-sorted cells, we removed 5 small contaminating clusters that expressed non-epithelial marker genes and contained many cells confidently predicted as non-epithelial by Seurat's Transfer Data' workflow with the Human Lung Cell Atlas reference[36]. For the other datasets, cluster annotation was aided by automated cell-type prediction using as reference epithelial cells from the Human Lung Cell Atlas. For the joint embedding of cells from lung tissue epithelial cells and lung organoids, we selected 2000 organoid cells and all epithelial cells from the reference and used Seurat's IntegrateData workflow.

Differential gene expression between the pool and HTII-280-derived organoids samples was analyzed using DESeq2 (version 1.30.1)[45] on aggregated "pseudobulk" counts for all cells from the same sample in a cluster, building a combined model for the protocol with donor identity as a covariate and "normal" shrinkage. For the IAV infection, we first built a model for the three factors donor + protocol + infection to assess gene expression changes between all infected and control samples. Next, we used a model for the combined factor protocol + infection with the donor as covariate, and tested infected vs. control cells in HT2 and pool organoid samples separately. Pathway analysis was performed using tmod[47] and the Hallmark, Reactome and Gene Ontology (BP) gene sets from MSigDB (version 7). Gene set scores for the ASCS signature were calculated using Seurat's AddModuleScore function. SCENIC analysis was run using default parameters, selecting the top 3 cluster-specific regulons. Monocle3 (version 0.2.3.0) was used for pseudotime ordering, using basal clusters as a starting point for trajectories. Transcriptome similarity between donors was assessed by using Seurat's AverageExpression function for each donor and coarse cell-type using the "integrated" assay followed by a principal component analysis. CNV analysis was performed using CONICS (version 0.0.0.1) (https://academic.oup.com/bioinformatics/article/34/18/3217/4979546) and entire chromosome arms as regions, splitting the dendrogram to get 2 clusters and designating the smaller one as potential CNV subclone.

**Statistics and reproducibility**. The study is based on an analysis of phenotypic and functional characteristics from total of 30 different donor tissues. A detailed overview is provided in Fig. 2b and Supplementary Data 2, which donors were used for which experiments. All figures and experiments include labels that identify donor samples used for a particular experiment. Statistical analysis and graphical representation of the data were performed by using Graph Pad Prism 9 software. The figure legends give full information about the number of independent biological replicates (*n*) analyzed and the type of statistical test, which was used to calculate significance. Statistical significance is presented as *$p < 0.05$, **$p < 0.01$, ***$p < 0.001$.

**Reporting summary**. Further information on research design is available in the Nature Research Reporting Summary linked to this article.

## Data availability

Processed sequencing data is deposited and available under GEO accession GSE197949. Lists of differentially regulated genes identified by the bioinformatic analysis are provided in Supplementary Data 1 and Supplementary Data 2 files. Source data underlying the graphs are provided in the Supplementary Data 3 file. Images of uncropped WB raw data are provided in Supplementary Fig. 7 . All other relevant data are available from the corresponding author on reasonable request.

## Materials availability

All materials and reagents will be made available upon the installment of the material transfer agreement (MTA).

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

## Acknowledgements

K.H., A.D.G., D.F., M.M., N.S., S.H., and A.C.H. are funded by the German Research Foundation (DFG) SFB-TR84; A.C.H., A.D.G. S.H., M.K., K.H., E.W., and M.L. are funded by the Federal Ministry of Education and Research (BMBF) project NUM-Organo-Strat (01KX2021); and A.D.G., M.K, K.H., M.B., A.C.H., and S.H. are funded by the Einstein Center 3R. A.C.H. and S.H. are funded by the Berlin Institute of Health (BIH), the Federal Ministry of Education and Research (BMBF) project RAPID as well as Charité 3R. A.C.H. is funded by Charité-Zeiss Center MultiDIM. E.W., L.G.T.A. and M.L. are funded by KA1-Co-02 COVIPA, a grant from the Helmholtz Association's Initiative and Networking Fund. The authors thank Doris Frey, and Katharina Hellwig, for technical help, Toralf Kaiser and Jenny Kirsch from Flow Cytometry & Cell Sorting DRFZ for FACS sorting and Sara Timm from the Core Facility for Electron Microscopy of the Charité for support in the preparation of samples for EM imaging. Computation has been performed on the HPC for the Research cluster of the Berlin Institute of Health. The stock of influenza virus Pan/99(H3N2) was kindly provided by Thorsten Wolff from Robert-Koch Institute Berlin.

## Author contributions

K.H., M.K., K.H.O., D.F., Z.D., A.L., M.M., and M.B. performed and analyzed experiments. L.G.T.A and E.W. performed the sequencing supervised by M.L. B.O. performed the analysis of sequencing data supervised by D.B. E.L.R. performed and analyzed EM imaging supervised by M.O. J.H., T.F. performed sample processing for immuno-fluorescence supervised by A.D.G. M.T., T.T.B., S.E., H.L.T., P.S., J.N., and J.C.R. provided lung tissue samples. A.C.H., S.H. and N.S. provided conceptual advice. M.K., K.H., and B.O. wrote the manuscript. M.K. conceived and supervised the study. All authors revised and approved the final version of the manuscript.

## Funding

## Competing interests

The authors declare no competing interests.
