## [Peer Review File · Communications Biology]

Reviewers' comments:

Reviewer #1 (Remarks to the Author):

Hoffmann et al assembled a cohort of ~15 (see Major Comment 3) patients undergoing surgical lobectomy of the lung for either a primary lung cancer, a secondary metastasis, or purulent inflammation and who mostly suffered from various co-morbidities. For each patient they obtained tissue from remote, apparently healthy regions, dissociated single cells, and sorted them based on immunostaining for HTII-280, an unknown marker of the apical membrane of Alveolar Epithelial Type 2 (AT2) cells. From 2 patients, they obtained expression profiles from 9,950 freshly dissociated, HTII-280 positive cells using 10x droplet-based capture. Most of these cells expressed AT2 cell markers (e.g. SFTPC), but some expressed markers of other cell types (AGER for AT1, SCGB1A1 for Club, and MUC5B for Goblet cells) and states (immediate early genes FOS, EGR1, JUN, and ZFP36 or proliferative genes STMN1, TYMS, HMGB2, and HMGN2).

The authors cultured unsorted and both HTII-280 positive and negative sorted cells isolated from nearly all of their patients with airway organoid media (AOM) or AOM supplemented with the Wnt pathway activator CHIR99201 (CHIR), known to be necessary for alveolar organoid growth. As expected, organoids from HTII-280 positive cells grown in AOM with CHIR had abundant AT2 marker immunostaining (e.g. SFTPC, HTII-280, and CDH1) and histological features (e.g. lamellar bodies). They dissociated cells in alveolar organoids from 3 patients and obtained 7,800 single cell expression profiles, which surprisingly showed most (~70-85%) had a basal-like or goblet-like cell identity, while a minority (~15-30%) had an AT2-like identity. 63,826 cells captured from organoids from 5 patients grown from HTII-280 positive cells without CHIR, by contrast, had only basal-like, goblet-like, and ciliated-like expression patterns. Most notably, they found knockdown of GSK3- β , a negative regulator of the Wnt pathway, was insufficient to robustly produce alveolar organoids with the same features as CHIR.

The authors also infected matched organoids made from unsorted cells cultured in AOM and HTII-280 positive cells cultured in AOM and CHIR from 3 patients as well as primary explanted tissue slices from 2 patients with the human influenza virus and measured the molecular and cellular response with single cell expression profiling on 55,146 and 12,419 cells, respectively. Productive infection and presence of influenza protein were confirmed by a plaque assay and immunostaining, also respectively.

In summary, the authors confirmed previous reports showing CHIR is a critical component of alveolar organoid media and began to unravel its potentially pleiotropic effects as well as confirm that organoids may be a useful model for respiratory viral infection. The last point has been demonstrated previously (two of many examples: Salahudeen et al Nature 2020 and Youk et al Cell Stem Cell 2020) and it is also unclear from the presented data that organoids are a superior model to explanted lung tissue. The primary novelty of this work comes from the observation that CHIR is apparently acting (see Major Comment 4 below) through an unknown, Wnt independent mechanism in driving alveolar organoid formation. This point is not followed up on and, instead, the authors perform a surface-level analysis of their expression data from infected organoids and explanted lung slices.

Major comments:

1. Why did the authors choose to focus on HTII-280 positive cells as opposed to AT2 cells as a whole or molecularly defined subsets of AT2 cells identified in human lung transcriptomic atlases?
2. The authors claimed they removed contaminant clusters from their single cell RNA sequencing dataset of freshly isolated, HTII-280 positive cells. How do they know other populations, such as the proliferative cluster they claim has a "stem cell signature" is not also a contaminant and/or an artifact of cell dissociation? The authors should perform additional immunostaining or single molecule fluorescence in situ hybridization (smFISH) to confirm these populations are HTII-280 positive in vivo.
3. If HTII-280 positive cells are not homogeneous, how would that effect the authors claim that they can enter diverging programs? An alternative interpretation could simply be that different culture conditions favor expansion of different cell types within the group.

4. The authors knocked down GSK3- β by about ~80% (Fig. 4g) as measured by qPCR. What percentage of cells are affected? How does this effect the level of GSK3- β protein present? How does this affect the expression of downstream Wnt target genes? It is not possible to evaluate the authors claim that CHIR has pleotropic, Wnt-independent effects without comparable data to Fig. 4e.
5. If CHIR's ability to drive alveolar organoid formation does turn out to be Wnt-independent, the authors should identify its relevant target. Why not perform the proposed experiments in lines 322-324?
6. Does the presence of PA, PB1, PB2, and NP actually confirm productive infection? This should be demonstrated by infecting different homogeneous cell types with influenza (some that are known to control the virus and some that cannot) and confirming detection of these viral genes (by expression profiling) serves a proxy for productive infection (through a plaque assay).

Minor comments:

1. More citations are needed throughout the manuscript (e.g. for claims on lines 63-64, 66-67 "firm lineage commitment", 120-121)
2. The authors should provide the gene list for the Adult Stem Cell Signature as a supplementary table (line 119)
7. It is unclear how the metadata presented in Fig 2b relates to the donors/organoids/culture conditions were used in experiments described in the manuscript. The authors should more clearly indicate which organoids (and how they were made and who they were made from). Of note, the manuscript states organoids grew from 14/16 patients (line 149), but Fig. 2b has only 15 patients.
3. The authors use cluster co-occupancy as a proxy for results being similar across organoids, but have applied batch correction to their dataset which could have smoothed biologically relevant axes of variation (e.g. in lines 256-8). The authors should quantify the similarity with another metric, such as correlation of pseudo-bulk expression profiles from molecularly defined populations across donors.
4. The effect of methanol fixation on the single cell expression profiles obtained should be explored/discussed. Table S2 is missing.
5. The methods for the knockdown of GSK3- β are missing.
6. It is unclear why the organoid model is superior to the lung tissue explant model for influenza infection. The datasets are not entirely comparable given different routes of viral delivery.
7. Organoids were only washed once after infection. How do the authors know the virions applied are adequately cleared?
8. There are a lot of immune cell doublets in the explant tissue infection dataset (Fig. S6d). How did the authors perform doublet discrimination and does this problem effect other datasets generated in this work?

Reviewer #2 (Remarks to the Author):

Summary comments:

In this manuscript, Hoffman, Obermayer et al. characterize HTII-280+ sorted cells and grow these alveolar progenitors in a previously published airway media (AOM) in the presence and absence of the GSK3 inhibitor CHIR99021. The major conclusions of the paper are: (1) CHIR99021 is required to maintain the alveolar lineage in AOM. (2) In the absence of CHIR99021, the AT2 cells are plastic and can differentiate into the airway lineage. (3) The effects of CHIR99021 are outside of its role as a GSK3B inhibitor. These observations are quite interesting and provocative but require some additional controls, experiments, and investigation before this work is suitable for publication.

Specific comments:

1. Based on the authors' data, CHIR seems to be required for alveolar progenitors to persist and proliferate in airway media. However, it is not clear if this is context-specific (e.g. dependent on the specific growth factors in the media, especially given some recent evidence regarding the role of EGF) or an absolute requirement. The authors do not need to completely redo their media conditions, but if they have any experiments that clarify whether an EGF-containing media still requires CHIR, that would be interesting to include.

2. The authors conclude that the alveolar progenitors isolated by sorting with HTII-280 selection can be plastic and exhibit airway differentiation in certain contexts. This is an interesting finding that perhaps may be supported by other emerging data in the field.

a. One alternative – low likelihood I realize but not impossible - explanation based on the way the authors have set up these experiments is that the sorting is not 100% pure and contamination by even a few stray basal cells could out-compete the AT2 cells, especially given the low colony formation efficiency after sorting, the fact that the media was originally developed to support airway cells and considering the months-long time that some of these experiments were performed which would allow for competition. The authors perform scRNA-seq in Fig. 1 which shows nearly universal expression of AT2 markers. However, it would also be good to characterize the immediate post-sort, HTII-280 negative sort population, serial passages by qPCR – This would also allow for an understanding of the kinetics of the plasticity. In general, the authors use Western blotting to examine markers, when qPCR would be more sensitive for airway markers and IF would allow for examination of morphology and frequency of cell events.

b. The plasticity argument would be further supported if the authors were able to perform some kind of trajectory/velocity analysis on their scRNAseq data in Figure 3E.

c. The authors should perform further analysis characterizing intermediates between AT2 -> airway differentiation. This further analysis could be highlighting markers of intermediates from the scRNAseq and validating expression by IF in organoid cultures.

d. In figure 4A, the authors should stain for additional markers beyond SCGB1A1 and DNA (e.g. AT2, basal markers). Similarly with figure 4D, the authors should also perform some IF.

3. The authors conclude that the effects of CHIR are outside of its role in Wnt activation given the evidence that the addition of CHIR does not further increase Wnt target transcription, GSK3B knockdown does not phenocopy CHIR treatment, and that CHIR is still required even in the presence of knockdown of GSK3B. This is a really interesting point if true, and so it behooves the authors to support this conclusion with additional controls and characterization.

a. The methods section in the document this reviewer received did not include any information about the shRNA experiment, including whether independent guides were used, the sequence of the guide, etc. There is no mention of a non-targeting guide (only 'control' organoids, which I presume are uninfected). There is evidence of GSK3beta knockdown in figure 4G, but given the known off-target effects with shRNA, at the very least I would have wanted to see at least 2 hairpins along with a non-targeting control, if not also a rescue construct.

b. The AOM contains R-spondin 1 in it. It would be interesting to know whether removal of R-spondin1 makes a difference in supporting alveolar organoid growth or plasticity in the presence and absence of CHIR, given the conclusion that the effects of CHIR are outside of its role as a Wnt activator. I would couple that experiment with a qPCR examining Wnt targets.

Reviewer #3 (Remarks to the Author):

SUMMARY:

This manuscript titled "Human alveolar progenitors generate dual lineage bronchioalveolar organoids" aimed to characterise alveolar progenitors in the human adult lung. These progenitors may act to replenish the alveolar compartment during cell turnover or tissue injury, but have not been extensively studied.

scRNA-seq of HTII-280+ cells from peripheral lung tissue revealed a distinct subcluster enriched for adult stem cell signature genes, as well as typical AT2 markers. The authors went on to show that organoids grown from HTII-280+ cells cultured in a stem cell-friendly medium could display alveolar or bronchial phenotypes depending on the respective presence or absence of GSK3-B inhibitor (and therefore Wnt signaling promotor) CHIR99201. This is in line with previous work showing CHIR99201 is required to promote alveolar differentiation in vitro.

However, these organoids did not express increased levels of a select few Wnt target genes, and using a lentiviral vector to silence GSK3-B in addition to CHIR did not increase AT2 marker expression. This could indicate that CHIR99201 acts pleiotropically to promote alveolar phenotypes, and Wnt signaling may only be part of the story.

Finally, the authors showed that HTII-280+ derived organoids can be infected with human influenza virus, and it infects both AT2 and bronchial cell types. These showed a transcriptomic response to infection that is comparable to lung tissue explants, indicating a good level of functional similarity with native cells.

OVERALL IMPRESSION:

My overall impression of the work is that it firstly represents a novel look into the subclustering of HTII-280+ cells in human adult lung tissue, and has implicated a subcluster which could be responsible for alveolar regeneration. However, there was no attempt to isolate this subcluster, and the subsequent work uses all HTII-280+ subtypes. I wonder if the authors looked into whether isolation of the subcluster by differential expression of genes is possible?

The study reaffirms previous work indicating that GSK3B inhibition by CHIR99201 appears important for suppression of bronchial lineage genes, and promotion of alveolar phenotypes in organoids. However the lack of certain Wnt target gene upregulation in these treated organoids indicates additional mechanisms of action by CHIR99201. This is a somewhat novel, important finding and would benefit from additional investigation to uncover which other CHIR99201 targets could be implicated, and would lend a lot of extra weight to this paper. Additionally or alternatively, inclusion of more Wnt target genes in the qPCR analysis is highly beneficial, as only three were shown in the analysis which does not rule out the importance of Wnt signaling entirely. Most crucially, the high levels of bronchial lineage cells (given by scRNAseq) in alveolar organoid cultures with CHIR99201 must be addressed, as this is not in line with previous work and could constitute a flaw in methodology as opposed to trans-differentiation of cells. The discussion suggests variation in other media components as a possible cause, and this would benefit from investigation by the authors – discovering that tweaks in media constituents leads to plasticity of HTII-280+ cells is an important finding if true, but the actual subsequent analyses presented in this paper would have benefitted from a purer alveolosphere population.

Ultimately I think the paper presents some interesting findings which are highly relevant in the field of lung repair and regeneration. However, some extra work or clarification is needed (see specific comments) in areas before definitive conclusions can be drawn.

SPECIFIC COMMENTS:

1. Abstract (p3, lines 57-58): rewrite to: "...lung organoids show a similar response to lung tissue explants, which confirms their suitability for studying the effects of pathogen-host interactions in the lung."

- did you mean sequelae? Don't think 'long-term' can quite be claimed yet.

2. Intro (p4, line 82): What is meant by "proposed mechanisms for the human lung"? Assuming this means in adulthood and not during development as there is evidence for non-unidirectional differentiation routes in humans, with type 1 and 2 progenitor commitment occurring earlier than expected. Please clarify.

3. Results (p5, line 109): Spelling – SFTPC

4. Results (p6, line 124): Spelling – transcription rather than transcriptional

5. Results (p6, line 125): Inconsistent italicisation – no need to just italicise TFs specifically, they are all genes

6. Results (p6, line 126): Confusing sentence structure - SOX2 and RARG not enriched in cluster 11, whereas TCF7L2 and HES4 are – presumably this is supposed to be implied by the word "partially" but does not obviously translate, particularly as the genes mentioned are all seemingly random. SOX2 not particularly enriched in 13 either – why mention it?

7. Results (p6, lines 135-136): Could it be variation of location of tissue received?

8. Figure 2: Table errors/questions:

- What is the LTC column?
- Why are some cells blank?
- Crossboxes in gastritis (D6), inflammation (D7), and transplantation (D26, also needs space)
- Need space between Typ2 (D3)
- Spelling: "sarcoidosis" (D5)
- Why not tested? (D6)
- semicolon > comma (D8)
- 1,6 > 1.6 (D10)
- Spelling & capitalise: "hypercholesterolaemia" (D24)
- What is 'non av' (D26)?

9. Figure 2: Is it necessary to have the HTII- population in C? Feels confusing and unnecessary - Focus should be on HTII+ in AOM vs AOM + CHIR. Dubious about including the non-sorted pool as well; both may be better as supplementary. Clear that formation efficacy is increased in non-sorted population but double the count of cells was seeded in the first place (1000 vs 2000 cells - methods). Also state scale bar dimension.

10. Results (p8, lines 190-192): 60 - 85% bronchial cell types (given by scRNAseq) in the alveolar organoids is concerning, and likely affecting subsequent data such as Wnt gene upregulation significance. Assuming a pure HTII-280 population was sorted initially, why are there so many bronchial lineage cells in the presence of CHIR? Are the AT2 and bronchial cells in separate, distinct organoids, or expressed together in one? If separate, Wnt gene analysis would be more robust if confined to pure alveolar organoids as opposed to the mixture.

What about staining for these bronchial markers or performing a protein-level analysis? This is partially addressed in Fig 4 with media constitution comparisons, but it feels very missing in this figure and Fig 4 does not answer the above concerns - I would add some staining images of KRT5/p63 or another Western as in 3(D) with airway cell markers to really assess the presence of bronchial cells in the AvO.

11. Figure 3 legend (p28, line 716): Can this be claimed when only a fraction are AT2 according to sequencing?

12. Figure 3 E: The subcluster numbers are never mentioned - what do they all refer to/ what are the key characteristics of each subcluster?

13. Figure 4 legend (p29, lines 735-739): D & E are the wrong way around on figure

14. Figure 4: (A) What about control images for dtyrTub?

(D) Why does TP63 have double bands for pre-sorted cells?

(F) Was any attempt made to analyse the shGSK3-B organoids further? It was claimed they had poor efficiency, but from the images some of them appear viable. Would be interesting to see their phenotype and whether Wnt target genes were upregulated here.

(G) This is not a phenotypic analysis as stated. Why is SCGB1A1 not significant? Would be good to have GSK3-B knockdown validated at protein level.

15. Results (p9), Figure 4D: How recently seeded were the organoids used for western blotting in 4(D)? Why do p5 AvO with CHIR express TP63 at low levels, when the same is not true for newly sorted AvO in fig S3(E)? And if pre-sorted AvO cells grown with CHIR express more or equal p63 than the newly sorted HTII-280 AvO organoids grown with or without CHIR, could the differences between the two conditions even out over time (before the next passage); i.e: are you confident the AOM alone condition would consistently express higher TP63 compared to AvO with CHIR?

Reviewer comments – Point-by-point response

Reviewer 1

Major comments:

1. Why did the authors choose to focus on HTII-280 positive cells as opposed to AT2 cells as a whole or molecularly defined subsets of AT2 cells identified in human lung transcriptomic atlases?

HTII-280 is a well-validated and robust surface marker specific for AT2 cells and has been routinely used by other groups to isolate AT2 cells (Gonzalez et al., Katsura et al., Youk et al., Kathiriya et al.). Moreover, our single-cell analysis of primary tissue isolates which we have now considerably expanded by the addition of four more donor tissues, confirms that the HTII-280⁺ sorted cells have an AT2 identity.

2. The authors claimed they removed contaminant clusters from their single-cell RNA sequencing dataset of freshly isolated, HTII-280 positive cells. How do they know other populations, such as the proliferative cluster they claim has a “stem cell signature” is not also a contaminant and/or an artifact of cell dissociation? The authors should perform additional immunostaining or single molecule fluorescence in situ hybridization (smFISH) to confirm these populations are HTII-280 positive in vivo.

We thank the reviewer for this constructive comment and suggestions. Realizing the importance of ensuring stringent experimental conditions for alveolar cell isolation we have added EpCAM as an additional marker and performed scRNA-seq of HTII-280⁺/EpCAM⁺ cells for 4 additional donors and thereby greatly increased the number of analyzed cells (33375). Single-cell RNA-seq data analysis resulted in very similar cluster composition and matching transcriptional profiles (Figure 1b and e and Supplementary figure 1c). After identifying cluster 11 as a candidate cluster to contain a progenitor population we have selected FOXM1 for validation based on localized expression (absence of expression in other clusters.) We have followed reviewers' suggestion and performed FOXM1/HTII-280 co-staining after Cytospin of HTII-280⁺ sorted cells and found nuclear FOXM1 expression in a subpopulation of the HTII-280⁺ cells (Fig. 1d) in agreement with single-cell sequencing data.

3. If HTII-280 positive cells are not homogeneous, how would that effect the authors claim that they can enter diverging programs? An alternative interpretation could simply be that different culture conditions favor expansion of different cell types within the group.

We show that although the cells can be divided into subclusters, all of them express the common AT2 marker and, by comparison with tissue data, are classified as AT2 type cells. The clusters therefore rather result from differences in differentiation and cell cycle status. It is well described in other tissues (e.g. Lgr5⁺ population in the intestinal tract) that only cells, which have progenitor potential in the native tissue, are capable of generating organoids *in vitro*. Though subpopulation of AT2 in the lung, which carries stemness potential remains incompletely understood, robust generation of alveolar organoids, which are expandable in long-term culture, as previously published by other groups and within this study, confirms their existence. And our sub-sorting experiments (Fig. 4c) of established alveolar lines exclude the possibility that airway properties originate from a separate pool of progenitors as these would have been out-competed in the initial organoid formation culture regarding the assumption of two different types of HTII-280 progenitors. Thus, we believe that our comprehensive data set of scRNA-seq analysis of HTII-280⁺ cells does provide important insight about the organization of the alveolar compartment and the putative stem cell population and could have great implications for future research in the field of lung regeneration and lung disease.

4. The authors knocked down GSK3- β by about ~80% (Fig. 4g) as measured by qPCR. What percentage of cells are affected? How does this effect the level of GSK3- β protein present? How does this affect the expression of downstream Wnt target genes? It is not possible to evaluate the authors claim that CHIR has pleotropic, Wnt-independent effects without comparable data to Fig. 4e.

The revised manuscript includes bulk RNA-seq data of GSK-3 β knockdown (KD) with two different shRNAs against *GSK-3 β* , which showed high knockdown efficiency. As organoids were subjected to puromycin selection after lentiviral transduction it can be assumed that only cells which contain the integrated vector survived. We did detect upregulation of downstream Wnt target genes, like LEF1 and WIF1, in *shGSK-3 β* organoids in comparison to control vector transduced line, confirming Wnt pathway activation (Fig. 5e). The data also indicate an additive effect of *GSK-3 β* KD and CHIR-treatment on Wnt target gene expression/Wnt pathway activation. However, our main aim was to find out which genes are regulated only by CHIR without being affected by KD of *GSK-3 β* and thus canonical Wnt pathway activation. Indeed, the bulk RNA-seq data yielded a number of genes affected only by the presence of CHIR in wildtype as well as *GSK-3 β* KD organoids. These genes include *FOXM1* and *EGF* (Fig. 5d and e).

5. If CHIR's ability to drive alveolar organoid formation does turn out to be Wnt-independent, the authors should identify its relevant target. Why not perform the proposed experiments in lines 322-324?

We thank the reviewer for this suggestion. As we also recognized the importance and implication of the pleiotropic effect of CHIR we have performed a global analysis of the transcriptional response of organoids in CHIR medium. By bulk RNA-seq of the CHIR-treated wildtype (non-mammalian shRNA) and *GSK-3 β* KD organoids, we identified genes only regulated by the presence/absence of CHIR, including *FOXM1* and *EGF*. As discussed, these two proteins are potentially important targets for the regulation of AT2 differentiation. Regarding the systemic comparison of alveolar and airway organoids from the same donors (Fig. 6e) the analysis clearly shows that despite the expression of specific markers of the airway and cell-type classification as "secretory and basal", cells in alveolar organoids still co-express considerable level of alveolar markers (*SFTPC*, *NAPSIN A*). This is in line with general morphological features of alveolar organoids which remain preserved as long as the organoids are cultivated in the CHIR medium.

6. Does the presence of PA, PB1, PB2, and NP actually confirm productive infection? This should be demonstrated by infecting different homogeneous cell types with influenza (some that are known to control the virus and some that cannot) and confirming detection of these viral genes (by expression profiling) serves a proxy for productive infection (through a plaque assay).

The study of Russel et al. (2018) in *E life* analyzed in detail the dynamics of Influenza infection on the single-cell level and presented evidence that expression of these viral transcripts is a useful tool to discriminate productivity of infection. Nevertheless, we would like to point out that we have accompanied each organoid infection experiment with plaque assay time course after infection (1h-120h post infection) (Fig. 7b) as a "gold standard" to measure viral replication. As infection experiments in our study were aimed as a proof of concept of the potential application of organoids, we felt more detailed analysis of the Influenza A infection, while certainly interesting, would be out of the scope of this project. Because our data clearly demonstrate potent viral infection and virus replication (IF images and plaque assay) in the organoid culture and the infection is not the main focus of the paper we did not include any further experiments here.

Minor comments:

1. More citations are needed throughout the manuscript (e.g. for claims on lines 63-64, 66-67 "firm lineage commitment", 120-121)

We added further citations as necessary.

2. The authors should provide the gene list for the Adult Stem Cell Signature as a supplementary table (line 119)

We included the gene list in Supplemental table 1. We have also provided a detailed list of all marker genes identified in clusters of HTII 280⁺/EpCAM⁺ cells.

7. It is unclear how the metadata presented in Fig 2b relates to the donors/organoids/culture conditions were used in experiments described in the manuscript. The authors should more clearly indicate which organoids (and how they were made and who they were made from). Of note, the manuscript states organoids grew from 14/16 patients (line 149), but Fig. 2b has only 15 patients.

We tried to improve information by indicating within the figure which donor was used in which experiment. In the table in Fig. 2b there are only the donors listed from which we retrieved HTII-280⁺ and HTII-280⁺/EpCAM⁺ sorted cells and from which either alveolar organoid lines grew or which were used in scRNA-seq experiments. Organoids were generated as described in the methods section.

3. The authors use cluster co-occupancy as a proxy for results being similar across organoids, but have applied batch correction to their dataset which could have smoothed biologically relevant axes of variation (e.g. in lines 256-8). The authors should quantify the similarity with another metric, such as correlation of pseudo-bulk expression profiles from molecularly defined populations across donors.

We thank the reviewer for this suggestion. We now compared not only the mixing of samples across the UMAP embedding but also checked the similarity of pseudobulk expression profiles in a PCA (see Supplementary fig. S3e and S4c).

4. The effect of methanol fixation on the single cell expression profiles obtained should be explored/discussed. Table S2 is missing.

Methanol fixation has been shown to preserve relevant transcriptional signals in single-cell analysis and does not affect data substantially. This is systematically investigated and published by Wang and colleagues (BMC genomics 2021). Also, the procedure is formally endorsed by the kit manufacturer (10x genomics) who provides a detail protocol which we have followed.

5. The methods for the knockdown of GSK3- β are missing.

We apologize for this omission in the first submission. A method section has been amended.

6. It is unclear why the organoid model is superior to the lung tissue explant model for influenza infection. The datasets are not entirely comparable given different routes of viral delivery.

The infection experiments in this study explored only the general comparability of two systems and we do show that organoids are suitable as a model to study Influenza A infection *in vitro* and complement the *ex vivo* lung tissue model. The obvious advantage of organoids would be the capability to design more long-term experiments, but also to perform infection in genetically engineered organoids. Organoids have also the key advantage of expandability, which enables high reproducibility.

7. Organoids were only washed once after infection. How do the authors know the virions applied are adequately cleared?

All infection experiments included control plaque assay which was performed 1h post-infection to account for any viral particles in the culture that did not infect organoid cells. At this time point no productive viral particles could be retrieved, therefore it can be concluded that all viruses detected at later time points originated by the productive infection cycle within the organoids.

8. There are a lot of immune cell doublets in the explant tissue infection dataset (Fig. S6d). How did the authors perform doublet discrimination and does this problem effect other datasets generated in this work?

We thank the reviewer for pointing out the potential influence of cell doublets. In the revised version, we added DoubletFinder to our data processing pipeline for all datasets.

Reviewer 2

1. Based on the authors' data, CHIR seems to be required for alveolar progenitors to persist and proliferate in airway media. However, it is not clear if this is context-specific (e.g. dependent on the specific growth factors in the media, especially given some recent evidence regarding the role of EGF) or an absolute requirement. The authors do not need to completely redo their media conditions, but if they have any experiments that clarify whether an EGF-containing media still requires CHIR, that would be interesting to include.

Indeed, as explained above we show now that EGF can enhance CHIR effect by further upregulating SFTPC expression (Fig. 3h), while we also find that CHIR induces EGF expression. However, we did not explore the effect of EGF in the absence CHIR, but the study of Ebisudani et al. (2021) strongly suggests that leaving out CHIR can not be complemented by EGF alone and requires a whole panel of additional growth factors. This we discuss in more detail in our revised manuscript. (lines 396-403)

2. The authors conclude that the alveolar progenitors isolated by sorting with HTII-280 selection can be plastic and exhibit airway differentiation in certain contexts. This is an interesting finding that perhaps may be supported by other emerging data in the field.

a. One alternative – low likelihood I realize but not impossible - explanation based on the way the authors have set up these experiments is that the sorting is not 100% pure and contamination by even a few stray basal cells could out-compete the AT2 cells, especially given the low colony formation efficiency after sorting, the fact that the media was originally developed to support airway cells and considering the monthslong time that some of these experiments were performed which would allow for competition. The authors perform scRNA-seq in Fig. 1 which shows nearly universal expression of AT2 markers. However, it would also be good to characterize the immediate post-sort, HTII-280 negative sort population, serial passages by qPCR – This would also allow for an understanding of the kinetics of the plasticity. In general, the authors use Western blotting to examine markers, when qPCR would be more sensitive for airway markers and IF would allow for examination of morphology and frequency of cell events.

For characterization of HTII-280 derived organoids and differentiation *in vitro* we apply characterization on protein (Western Blot and IF) as well as RNA (qPCR and scRNA-seq) level. In Supplementary figure 3b we performed qPCR for AT2 and airway marker in different passages of organoid culture.

Including 4 more samples in the scRNA-seq data also revealed no presence of any stray basal cells but further strengthened the use of HTII-280 as a highly specific AT2 marker. Also, the use of peripheral lung tissue already limits the possible contamination with airway basal cells rather we encountered some slight contamination with immune cells which could be identified in the single-cell data as separate clusters and were thus eliminated. Moreover, as given in Supplementary figure 2c expression of alveolar as well as bronchial markers does not notably change over several passages, indicating constant differentiation of both cell lineages. Nevertheless, the frequency of alveolar versus bronchial differentiation in one culture seems to be donor specific, thus depending on the genetic and pathological background.

b. The plasticity argument would be further supported if the authors were able to perform some kind of trajectory/velocity analysis on their scRNAseq data in Figure 3E.

We now performed monocle 3 pseudotime analysis. The analysis revealed the differentiation route of less differentiated progenitors towards the fully differentiated airway and AT2 cell types. Results are provided in Figure 6f.

c. The authors should perform further analysis characterizing intermediates between AT2 -> airway differentiation. This further analysis could be highlighting markers of intermediates from the scRNAseq and validating expression by IF in organoid cultures.

Principal component analysis of the scRNA-seq data of our alveolar organoids revealed that no AT2 cell clusters diverge more widely in transcriptional profiles (see Supplementary figure 3e) and individual donor characteristics appear to have a dominant influence. Having in mind the novel study of Kathirya et al. (Nature Cell Biology, 2021) showing the potential of mesenchyme from pulmonary fibrosis patients to promote transdifferentiation of alveolar progenitors to basal cell types, it is tempting to speculate that these differences could be associated with the different clinical background of our donors.

d. In figure 4A, the authors should stain for additional markers beyond SCBG1A1 and DNA (e.g. AT2, basal markers). Similarly with figure 4D, the authors should also perform some IF.

To fully illustrate the shift in the phenotype which is driven solely by the medium change we have now included a panel of images from two organoid lines where organoids are grown in AOM and CHIR medium in parallel (Fig. 4a). While SCBG1A1 can be detected in alveolar organoids in the CHIR medium, levels dramatically rise in AOM medium. Importantly, SFTPC is basically only present in CHIR medium, reiterating the importance of the inhibitor for the induction of the alveolar differentiation.

3. The authors conclude that the effects of CHIR are outside of its role in Wnt activation given the evidence that the addition of CHIR does not further increase Wnt target transcription, GSK3B knockdown does not phenocopy CHIR treatment, and that CHIR is still required even in the presence of knockdown of GSK3B. This is a really interesting point if true, and so it behooves the authors to support this conclusion with additional controls and characterization.

a. The methods section in the document this reviewer received did not include any information about the shRNA experiment, including whether independent guides were used, the sequence of the guide, etc. There is no mention of a non-targeting guide (only 'control' organoids, which I presume are uninfected). There is evidence of GSK-3 β knockdown in figure 4G, but given the known off-target effects with shRNA, at the very least I would have wanted to see at least 2 hairpins along with a non-targeting control, if not also a rescue construct.

We have tested the influence of recombinant EGF on alveolar differentiation and found that it causes a further increase in SFTPC expression. Data is now included in Fig. 3h. It is clear that EGF-containing medium still needs CHIR for alveolar induction as protocols published by Youk. et al. and Katsura et al. do use CHIR99021 and EGF. We do however find that EGF is regulated by CHIR99021 among differentially regulated genes that are not affected by GSK-3 β knockdown. Because of the central importance of the medium composition for understanding the biology of the alveolar organoids, this is now discussed in more details outlined above.

In the methods section, a passage about the generation of knockdown organoid lines was added including specific information regarding used hairpins (lines 602-605). Indeed, we already used a control shRNA with a non-mammalian target sequence in the first version of the study. Moreover, we now improved the experimental setup by adding a second GSK-3 β targeting shRNA and then performed bulk RNA-sequencing of +/-CHIR treated knockdown and wild-type (control shRNA) organoid lines. Both shRNA sequences resulted in similar knockdown efficiency.

b. The AOM contains R-spondin 1 in it. It would be interesting to know whether removal of R-spondin1 makes a difference in supporting alveolar organoid growth or plasticity in the presence and absence of CHIR, given the conclusion that the effects of CHIR are outside of its role as a Wnt activator. I would couple that experiment with a qPCR examining Wnt targets.

It is clear that CHIR is essential for the alveolar induction, and as a Wnt activating agent,, it mimics the action of RSPO1. Based on this, it was a general consensus that the requirement for CHIR addition means that alveolar progenitors depend on higher Wnt activity. Before use in the organoid medium, the conditioned Rspo1 supernatants are tested on a Wnt reporter cell line to confirm its activity. So, assuming that the addition of Rspo1 to the organoid culture already induces canonical Wnt signalling,

this would further confirm that CHIR mode of action in alveolar differentiation is not primarily due to Wnt pathway activation. Ebisudani et al. (Cell reports, 2021) recently tested and demonstrated that Wnt agonists are necessary for the formation and expansion of alveolar organoids, so we have focused on additional genes which are regulated by CHIR that have by now have not been investigated.

1. Abstract (p3, lines 57-58): rewrite to: "...lung organoids show a similar response to lung tissue explants, which confirms their suitability for studying the effects of pathogen-host interactions in the lung." - did you mean sequelae? Don't think 'long-term' can quite be claimed yet.

We adapted this accordingly in the abstract.

2. Intro (p4. line 82): What is meant by "proposed mechanisms for the human lung"? Assuming this means in adulthood and not during development as there is evidence for non-unidirectional differentiation routes in humans, with type 1 and 2 progenitor commitment occurring earlier than expected. Please clarify.

This was now clarified. It reads now "By now proposed mechanisms of the regulation of the regeneration potential in the adult human lung" line 76

3. Results (p5, line 109): Spelling – SFTPC

Amended.

4. Results (p6, line 124): Spelling – transcription rather than transcriptional

Amended.

5. Results (p6, line 125): Inconsistent italicisation – no need to just italicise TFs specifically, they are all genes

Amended.

6. Results (p6, line 126): Confusing sentence structure - SOX2 and RARG not enriched in cluster 11, whereas TCF7L2 and HES4 are – presumably this is supposed to be implied by the word "partially" but does not obviously translate, particularly as the genes mentioned are all seemingly random. SOX2 not particularly enriched in 13 either – why mention it?

This paragraph is substantially revised, as new data sets were included in the analysis.

7. Results (p6, lines 135-136): Could it be variation of location of tissue received?

We agree with the reviewer's remark that precise localisation of the sample could also influence the number of HTII-280⁺ cells. However, this is a confounding factor that cannot be properly investigated without serial sampling from different localisation of the same lung which is a procedure that could not be implemented in current clinical practice due to its invasive nature.

8. Figure 2: Table errors/questions:

- What is the LTC column?
- Why are some cells blank?
- Crossboxes in gastritis (D6), inflammation (D7), and transplantation (D26, also needs space)
- Need space between Typ2 (D3)
- Spelling: "sarcoidosis" (D5)
- Why not tested? (D6)
- semicolon > comma (D8)
- 1,6 > 1.6 (D10)
- Spelling & capitalise: "hypercholesterolaemia" (D24)
- What is 'non av' (D26)?

The table has been revised accordingly. We did not use all donor samples for experimental applications. For example, for donor 6 all HTII-280 cells were used up for single-cell sequencing and no organoids were generated.

9. Figure 2: Is it necessary to have the HTII- population in C? Feels confusing and unnecessary - Focus should be on HTII+ in AOM vs AOM + CHIR. Dubious about including the non-sorted pool as well; both may be better as supplementary. Clear that formation efficacy is increased in non-sorted population but double the count of cells was seeded in the first place (1000 vs 2000 cells - methods). Also state scale bar dimension.

We do find it important to include non-sorted pool to understand the context of airway progenitors which outcompete alveolar progenitors even in the presence of CHIR (which we systematically test and show in Figures 6a and c). Thus, there is a necessity to separate HTII-280⁺ cells immediately in order to generate alveolar organoids. To underline this point we have now performed also organoid forming experiments with HTII-280⁺/EpCAM⁺ progenitors in AOM medium and compared them with HTII-280⁻/EpCAM⁺ (airway progenitors) and found again that HTII-280⁺ cells do generate small airway organoids but these are smaller in size and grow more slowly than (HTII-280⁻/EpCAM⁺) (Fig. 2e new).

10. Results (p8, lines 190-192): 60 - 85% bronchial cell types (given by scRNAseq) in the alveolar organoids is concerning, and likely affecting subsequent data such as Wnt gene upregulation significance. Assuming a pure HTII-280 population was sorted initially, why are there so many bronchial lineage cells in the presence of CHIR? Are the AT2 and bronchial cells in separate, distinct organoids, or expressed together in one? If separate, Wnt gene analysis would be more robust if confined to pure alveolar organoids as opposed to the mixture.

What about staining for these bronchial markers or performing a protein-level analysis? This is partially addressed in Fig 4 with media constitution comparisons, but it feels very missing in this figure and Fig 4 does not answer the above concerns - I would add some staining images of KRT5/p63 or another Western as in 3(D) with airway cell markers to really assess the presence of bronchial cells in the AvO.

As already discussed above, and now highlighted in the text (lines 220-228) we do believe that the more prominent bronchial features of our alveolar organoids, than published in studies with overall similar protocols (Youk et al, Katsura et al), are related to the differences in the medium composition and absence of recombinant EGF and MAP kinase inhibitor. We did perform additional experiments and could validate the positive effect of EGF on SFTPC expression (Fig. 3h). Also, analysis in Fig. 6e does confirm that cells classified as airway (basal and secretory) in alveolar organoids still express alveolar lineage markers, and we have true mixed phenotypes not a mix of alveolar and bronchial organoids. This is further supported by imaging of organoids (Fig. 4a).

11. Figure 3 legend (p28, line 716): Can this be claimed when only a fraction are AT2 according to sequencing?

Yes, we do believe that our study brought additional added value as we closely define conditions for sustained maintenance of the AT2 cells in long-term culture. Importantly, principal efficacy of the induction in CHIR medium is exceptionally high (15/17 donors tested). Also, we have performed the majority of our experiments including scRNA-seq in advanced cultures (> 3 months cultivated), which underlines progenitor potential, but could be the additional factor in the diversification of the phenotypes *in vitro*.

12. Figure 3 E: The subcluster numbers are never mentioned - what do they all refer to/ what are the key characteristics of each subcluster?

Based on the RNA profiles the cells can be assigned to different clusters. However, our major focus here was on cell type identification and we did not get into further characterization of the different subclusters beyond testing for stem cell signature. However, this would be very interesting in further studies. Monocle 3 pseudotime analysis (Fig. 6f) indicates that the clusters occur due to differentiation status (brighter colour towards fully differentiated cells).

13. Figure 4 legend (p29, lines 735-739): D & E are the wrong way around on figure
This issue is now amended.

14. Figure 4: (A) What about control images for dtyrTub?

Figure 4a was revised and control images for dtyrTub in AOM+CHIR have been added.

(D) Why does TP63 have double bands for pre-sorted cells?

We don't have an exact explanation for this phenomenon, but it could likely be the consequence of differentiation stages during very prolonged cultivation and potential modifications of TP63 protein in some cells. Both organoid lines (Fig. 4d) were in culture for 5 and 3,5 months, respectively, at the time of the sub-sorting experiment.

(F) Was any attempt made to analyse the shGSK3-B organoids further? It was claimed they had poor efficiency, but from the images some of them appear viable. Would be interesting to see their phenotype and whether Wnt target genes were upregulated here.

Yes, as this issue was raised by all reviewers as explained above we did extend our study of shGSK-3 β organoids and generated an additional knockout line with a second hairpin. While some organoids in AOM appear viable in phase-contrast images it was impossible to expand them and propagate them. Thus, we concluded that their progenitor potential is abolished. To circumvent this problem we have now designed the experiment (Fig. 5a) where organoids are grown in CHIR medium, allowed to form, and then inhibitor was withdrawn. New data is presented in detail in lines 269-289.

(G) This is not a phenotypic analysis as stated. Why is SCGB1A1 not significant? Would be good to have GSK3-B knockdown validated at protein level.

SCGB1A1 is not significant because of the variance between the donors. Experiment did confirm that the expression level of SCGB1A1 is relatively low in alveolar organoids (Fig. 5c). We do provide now supplementary table showing differential gene expression for all genes between alveolar and airway organoids to allow for complete phenotypic assessment (see Supplementary table 2).

15. Results (p9), Figure 4D: How recently seeded were the organoids used for western blotting in 4(D)? Why do p5 AvO with CHIR express TP63 at low levels, when the same is not true for newly sorted AvO in fig S3(E)? And if pre-sorted AvO cells grown with CHIR express more or equal p63 than the newly sorted HTII-280 AvO organoids grown with or without CHIR, could the differences between the two conditions even out over time (before the next passage); i.e: are you confident the AOM alone condition would consistently express higher TP63 compared to AvO with CHIR?

Yes, we are confident that organoids will always express a higher level of TP63 in AOM medium. As discussed above we do believe that long-term expansion > 3-4 months does cause additional drift and promotes the expression of airway markers. However, this is reversible, and this is also a message of critical importance for the overall interpretation of our data and their potential implications. We do discuss this issue in detail (Lines 412-419).

Reviewers' comments:

Reviewer #3 (Remarks to the Author):

The authors have put a lot of effort into this revision, improved the manuscript significantly and tried to address all the reviewers' concerns as much as possible. I still wonder how much of this work is actually novel based on what is already published, although their finding that the effects of CHIR99021 are outside of its role as a GSK3B inhibitor, is indeed interesting.

Reviewer #4 (Remarks to the Author):

Remaining comments

Reviewer 1 (original numbering maintained):

Major comments

2. The reviewer asks for validation on intact sectioned tissue. If I read their response correctly they did not perform this, but added extra markers and selections on dissociated cell samples. Increases the credibility but it doesn't properly exclude an artefact.

3. The assertion that a second subpopulation will be outcompeted is not formally demonstrated and it should not be presented as an unquestionable fact. The authors should present the scenario advanced by R1 and their view in the discussion

4. The authors only reply to the second part (Wnt target gene regulation). They do not show if the 20% GSK3b expression comes from 20% escaping cells, or all cells express a bit of GSK3b (easily done). This is an important point and their assumption was not validated. I would insist they respond in other way than just using omics, especially for questions that involve positional values.

5. The authors detected genes potentially regulated by CHIR but further validation should be shown (but not absolutely necessary in this manuscript). They can add in discussion that further validation is required.

Minor comments

8. The authors should discuss the effect of the presented numbers on their conclusions. I'd ask them to comment on their doublet numbers and if these affect their conclusions (discussions or results), Even if the numbers are actually not impacting on the conclusions, this should be stated.

Reviewer 2 (original numbering):

1. In the context described by R2 the EGF effect in the absence of CHIR should have been properly explored and I see no reason why not. If I'd be generous I'd allow it to pass but probably ask them to add this drawback (not doing this experiment and the reviewer's potential scenario) in the discussion.

2. a. This comment is similar with the one of R1. The authors have not conducted further key experiments but should at least cover the scenario proposed by R2 and their own take on it in the discussion

d. The authors have not responded to R2's request for extra markers, refused to include IF staining and also did not perform IS for Fig4d. This is key.

3. b. Not responded to this suggestion which would have added value to the paper (but perhaps not compulsory)

Reviewer 3 (original numbering):

10. Validation is required and the authors should perform the experiments in the second paragraph as, once more this is an important concern. The IF required must be performed.

14. (g) Again no validation at protein level (necessary).

Reviewers' comments:

Reviewer #3 (Remarks to the Author):

The authors have put a lot of effort into this revision, improved the manuscript significantly and tried to address all the reviewers' concerns as much as possible. I still wonder how much of this work is actually novel based on what is already published, although their finding that the effects of CHIR99021 are outside of its role as a GSK3B inhibitor, is indeed interesting.

We appreciate the acknowledgment of our efforts and new data and thank the reviewer for constructive comments that help improve our study further.

Reviewer #4 (Remarks to the Author):

Remaining comments

Reviewer 1 (original numbering maintained):

Major comments

2. The reviewer asks for validation on intact sectioned tissue. If I read their response correctly they did not perform this but added extra markers and selections on dissociated cell samples. Increases the credibility but it doesn't properly exclude an artefact.

We have now performed new stainings in native lung tissue sections from two different donors and included images in Fig. 1g and identified the presence of actively proliferating alveolar cells as well as AT2 cells expressing FOXM1 transcription factor in agreement with scRNA-seq and cytospin data.

3. The assertion that a second subpopulation will be outcompeted is not formally demonstrated and it should not be presented as an unquestionable fact. The authors should present the scenario advanced by R1 and their view in the discussion.

We thank the reviewer for the suggestion and have now included more nuanced language in the discussion (See Lines 426-429).

4. The authors only reply to the second part (Wnt target gene regulation). They do not show if the 20% GSK3b expression comes from 20% escaping cells, or all cells express a bit of GSK3b (easily done). This is an important point and their assumption was not validated. I would insist they respond in other way than just using omics, especially for questions that involve positional values.

As explained in the materials and methods section, lentiviral plasmids carrying shRNA also have puromycin cassette, and selection has been performed accordingly thus it is very unlikely that any untransduced cells escaped. We have now included Western blot (Supplemental figure 3g) of the shGSK3 β organoid line in comparison to control and the quantification by densitometry confirming knockdown efficiency of 80 %.

5. The authors detected genes potentially regulated by CHIR but further validation should be shown (but not absolutely necessary in this manuscript). They can add in discussion that further validation is required.

We have now added clarifying statements in the discussion regarding future direction (Lines 448-451)

Minor comments

8. The authors should discuss the effect of the presented numbers on their conclusions. I'd ask them to comment on their doublet numbers and if these affect their conclusions (discussions or results), Even if the numbers are actually not impacting on the conclusions, this should be stated.

The Materials and Methods section includes an explanation that doublets have been removed by the Doublet finder and thus don't affect the analysis. (Line 694)

Reviewer 2 (original numbering):

1. In the context described by R2 the EGF effect in the absence of CHIR should have been properly explored and I see no reason why not. If I'd be generous I'd allow it to pass but probably ask them to add this drawback (not doing this experiment and the reviewer's potential scenario) in the discussion.

We thank the reviewer for the comment but would like to point out that we did include new experiments and analysis of the EGF effect on the organoid differentiation in the revised manuscript in response to the original suggestion. Fig 3h illustrates strong induction in SFTPC expression quantified by qPCR in response to the presence of 50 ng recombinant EGF in the medium in comparison to the standard AO medium we have used in this study. Also, we discuss in detail the role of EGF in the discussion section (Lines 401-410).

2. a. This comment is similar with the one of R1. The authors have not conducted further key experiments but should at least cover the scenario proposed by R2 and their own take on it in the discussion.

As explained above we have now provided new comprehensive immunofluorescence evidence to support the study conclusions. In addition, we have added a more nuanced angle to the discussion to point out the remaining open questions (Lines 426-429).

d. The authors have not responded to R2's request for extra markers, refused to include IF staining and also did not perform IS for Fig4d. This is key.

Our revised version did include more IF stainings and demonstrated a difference in the differentiation of the organoids (SFTPC marker AT2 Figure 4A) depending on the medium (+/-Chir). However, recognizing the importance of the findings of mixed phenotypes in our sc RNA dataset and the requirement to confirm the dual differentiation potential of the HTII280 progenitors by independent experimental readouts, we have now performed additional IF stainings combining HT280 alveolar marker with TP63 marker of airway stem cells as well as dual staining of SFTPC with airway marker KRT5. As clearly shown in new Fig_6g organoids our alveolar organoids do contain groups of cells that co-express both classes of differentiation markers confirming dual phenotypes.

3. b. Not responded to this suggestion which would have added value to the paper (but perhaps not compulsory)

We do address the role of Wnt signaling in the discussion in the context of also findings from other studies (Ebisudani et al 2021) and our data but felt that experimental follow-up in this direction was not necessary for the main conclusions of our study (see lines 436-441) which point to important

Wnt independent effects of CHIR99021 inhibitor for the alveolar differentiation.

Reviewer 3 (original numbering):

10. Validation is required and the authors should perform the experiments in the second paragraph as, once more this is an important concern. The IF required must be performed.

In line with the comments of Reviewer 2 and suggestions of Reviewer 4, as explained above in detail we do now provide IF evidence and confocal imaging of the dual alveolar/ airway phenotypes in the HTII280+ derived organoids.

14. (g) Again no validation at protein level (necessary).

As indicated above study now does include WB validation of the knockdown efficiency on the protein level (Figure suppl. 3 g)

REVIEWERS' COMMENTS:

Reviewer #4 (Remarks to the Author):

The authors made additional efforts to reply to the remaining issues. In my opinion, most of the comments are now addressed. I have no additional comments.